# CSRv2: Unlocking Ultra-Sparse Embeddings

**Lixuan Guo**[1,2*]  **Yifei Wang**[3*]  **Tiansheng Wen**[1,2*]  **Yifan Wang**[1]  **Aosong Feng**[6]
**Bo Chen**[2]  **Stefanie Jegelka**[4,5]  **Chenyu You**[1†]
[1] Stony Brook University  [2] Xidian University  [3] Amazon AGI SF Lab[‡]  [4] TUM
[5] MIT  [6] Yale University

## ABSTRACT

In the era of large foundation models, the quality of embeddings has become a central determinant of downstream task performance and overall system capability. Yet widely used dense embeddings are often extremely high-dimensional (e.g., 4096), incurring substantial costs in storage, memory, and inference latency. To address these, Contrastive Sparse Representation (CSR) is recently proposed as a promising direction, mapping dense embeddings into high-dimensional but $k$-sparse vectors, in contrast to compact dense embeddings such as Matryoshka Representation Learning (MRL). Despite its promise, CSR suffers severe degradation in the ultra-sparse regime (e.g., $k \leq 4$), where over 80% of neurons remain inactive, leaving much of its efficiency potential unrealized. In this paper, we introduce **CSRv2**, a principled training approach designed to make ultra-sparse embeddings viable. CSRv2 stabilizes sparsity learning through progressive $k$-annealing, enhances representational quality via supervised contrastive objectives, and ensures end-to-end adaptability with full backbone finetuning. CSRv2 reduces dead neurons from 80% to 20% and delivers a 14% accuracy gain at $k = 2$, bringing ultra-sparse embeddings on par with CSR at $k = 8$ and MRL at 32 dimensions, *all with only two active features*. While maintaining comparable performance, CSRv2 delivers a $7\times$ speedup over MRL, and yields up to **$300\times$ improvements in compute and memory efficiency** relative to dense embeddings in e5-mistral-7b-instruct-based text representation. Extensive experiments across text (MTEB, multiple state-of-the-art LLM embeddings (Qwen and e5-Mistral-7B), SPLADEv3, GraphRAG) and vision (ImageNet-1k) demonstrate that CSRv2 makes ultra-sparse embeddings practical without compromising performance, where CSRv2 achieves 7%/4% improvement over CSR when $k = 4$ and further increases this gap to 14%/6% when $k = 2$ in text/vision representation. By making extreme sparsity viable, CSRv2 broadens the design space for large-scale, real-time, and edge-deployable AI systems where both embedding quality and efficiency are critical. Code is available at https://github.com/Y-Research-SBU/CSRv2.

## 1 INTRODUCTION

In the era of large foundation models, the quality of embeddings has become a decisive factor shaping downstream performance across tasks such as retrieval, classification and recommendation. Yet the dominant practice still relies on dense representations with thousands of dimensions (e.g., 2048 – 8192). While highly expressive, such embeddings incur substantial costs in storage, memory, and inference latency. These inefficiencies are magnified in large-scale and real-time deployments, where embedding computation and serving often dominate system throughput. As models scale further, embedding efficiency emerges as a central bottleneck, which limits both web-scale applications and deployment on resource-constrained platforms such as mobile and edge devices.

Several methods have been proposed to improve embedding efficiency, but they face sharp trade-offs under extreme compression. Existing approaches improve efficiency but falter under extreme com-

---

[*]Equal Contribution.

[†]Corresponding author: `chenyu.you@stonybrook.edu`

[‡]This work was done at MIT prior to Yifei Wang joining Amazon.

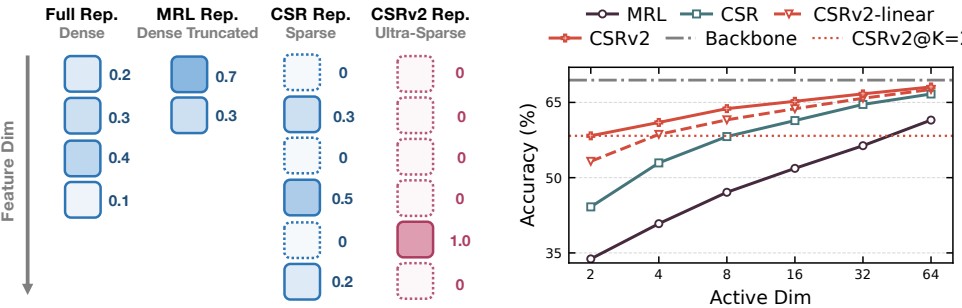

(a) Different efficient embedding schemes.  (b) Average accuracy on text embedding tasks.

Figure 1: **Overview of our proposed method. (Left)**: An illustrative comparison between full embedding, truncated MRL embedding, medium-sparse CSR embedding and ultra-sparse CSRv2 embedding. **(Right)**: Comparison of average text embedding performance on 6 types of tasks in MTEB benchmark with e5-mistral-7b-instruct backbone. To ensure a fair comparison, all methods are trained on the same data. We refer to the e5-mistral-7b-instruct model without task-specific finetuning as the 'Backbone' baseline.

pression. Matryoshka Representation Learning (MRL) (Kusupati et al., 2022) trains embeddings to function at multiple truncation lengths, yet expressivity collapses and accuracy drops sharply below a hundred dimensions. Contrastive Sparse Representation (CSR) (Wen et al., 2025) instead maps embeddings into high-dimensional sparse vectors, outperforming MRL and matching its quality with only one-quarter of the active dimensions. Despite this potential, CSR **suffers severe degradation in the ultra-sparse regime** ($k = 2$ or $4$). We refer to this regime as *ultra-sparse embeddings*, which in principle can deliver over $100\times$ efficiency gains in large-scale retrieval. However, existing methods incur $20 - 40\%$ accuracy losses in this regime, rendering such embeddings impractical in real-world scenarios. This raises a central question:

> *Are ultra-sparse embeddings inherently constrained, or can proper training mitigate this?*

Driven by this question, we take a closer look at ultra-sparse embeddings and identify three key challenges. First, they suffer from a "massive dead neuron" problem: even with modern mitigation techniques, more than 85% of neurons remain permanently inactive when CSR activates only two neurons ($k = 2$), severely limiting expressivity. Second, the mismatch between pretraining objectives and downstream tasks becomes amplified under ultra-sparsity, so CSR relying on purely self-supervised signals (e.g., image cropping) leads to pronounced degradation. Third, we observe that CSR also shows greater degradation when jointly trained on multiple datasets and domains, indicating that restricting it to a linear layer on top provides insufficient representational capacity.

To address the above challenges, we develop CSRv2, an improved training recipe for sparse embeddings that is as simple and generic as CSR(v1) yet delivers substantial and consistent gains in ultra-sparse regimes. CSRv2 combines a curriculum annealing schedule, which prevents early collapse when learning ultra-sparse embeddings, with natural supervision from labeled data, which replaces the noisy self-supervision of CSR and utilizes the few active dimensions more effectively. In addition, beyond training only a linear layer (CSRv2-linear), we explore finetuning the entire backbone with our objectives, analogous to the MRL setting, and show that this further improves generalization across domains, establishing new state-of-the-art results and outperforming MRL by up to 25% under the same training conditions. Altogether, CSRv2 provides the first reliable recipe for shrinking modern embeddings to just two or four active dimensions with only modest performance drops. This opens a new understanding of representational capacity and paves the way for extremely memory- and compute-efficient applications such as edge devices, robotics, and real-time search engines. We discuss in detail the evolution of text embedding and adaptive embedding techniques in Appendix A, highlighting the correlations and limitations of existing methods that motivate CSRv2.

To summarize, our contributions are:

Table 1: Overview of the training paradigms, objectives, trainable parameters, and performance (cf. Figure 1b) of the four efficient embedding methods discussed in this paper.

| Method | Training | Objectives | Trainable Params |
|--------|----------|-----------|------------------|
| MRL | Supervised | Multi-length Cross Entropy | Full Finetuning |
| CSR | Self-supervised | SAE + Contrastive | Linear Head |
| **CSRv2-linear** | Self-sup. + Sup. | $k$-annealing SAE + Sup. Contrastive | Linear Head |
| **CSRv2** | Self-sup. + Sup. | $k$-annealing SAE + Sup. Contrastive | Full Finetuning |

- We systematically explore the regime of ultra-sparse embeddings and diagnose three main causes of failure in prior methods: dead neurons, lack of effective supervision, and limited model capacity.
- We propose CSRv2, a simple and generic training recipe that addresses these issues through $k$-annealing for ultra-sparsity, supervised sparse contrastive learning, and optional full-model finetuning for multi-domain robustness.
- We validate CSRv2 extensively on text (six MTEB tasks and two domains in GraphRAG-Benchmark) and image (ImageNet-1k), show up to $4\times$ efficiency gains over CSR and $16\times$ over MRL at comparable performance, and attain $10\% - 30\%$ accuracy improvements on state-of-the-art Qwen3 Embedding models under short embedding lengths.

Our training data, code, and CSRv2-enhanced versions of Qwen3 and e5-Mistral-7B are available on https://github.com/Y-Research-SBU/CSRv2, ensuring compatibility with existing model configurations and readiness for production use. We are further committed to extending CSRv2 to a broader set of open-source models. By releasing these resources, we aim to encourage new research directions and practical applications of ultra-sparse embeddings that have not yet been explored.

## 2 BACKGROUND

The goal of representation learning is to map high-dimensional inputs (such as images or text) into low-dimensional embeddings that capture semantic similarity. Consider text embeddings as an example: given a batch of query–document pairs that share similar semantics, an LLM backbone encodes them into embedding pairs $\{(q_1, d_1), \ldots, (q_N, d_N)\}$, where $(q_i, d_i)$ denotes a query–document pair. The embeddings are then trained with a contrastive loss such as InfoNCE (Oord et al., 2018). However, standard embeddings typically remain high-dimensional (2k–8k), creating a significant bottleneck for large-scale, real-time retrieval systems, including search, recommendation, and retrieval-augmented generation. Here, we review two representative approaches to address this by producing embeddings with adaptive dimensionality for efficient applications.

**Matryoshka Representation Learning (MRL).** Instead of applying the loss solely on the full-size embeddings, MRL (Kusupati et al., 2022) truncates the first $m \in \mathcal{M}$ dimensions of the text embeddings $d[1:m] \in \mathbb{R}^m$ and applies the same loss function on a set of truncated lengths $\mathcal{M}$ with relative importance scale $c_m$. Formally, the objective of MRL is as follows:

$$\mathcal{L}_{\text{MRL}} = -\frac{1}{N} \sum_{m \in \mathcal{M}} c_m \sum_{i \in [N]} \log \frac{\exp\left(s(q_{i\,1:m}, d_{i\,1:m})/\tau\right)}{Z_i} \tag{1}$$

where $s(\cdot, \cdot)$ is the similarity function (cosine similarity in most cases), $\tau$ is the temperature parameter and $Z_i$ denotes the normalization factor that comes in different forms (Zhang et al., 2025a;b). Generally, the number of selected truncating lengths $|\mathcal{M}|$ will not be larger than $\lfloor \log(d) \rfloor$ of the original embedding size $d$ and all the relative importance scale $c_m$ will be set to 1.

**Contrastive Sparse Representation (CSR).** Instead of training the whole model as in MRL, CSR (Wen et al., 2025) takes a pretrained encoding model (with frozen weights), and trains a simple sparse autoencoder (Cunningham et al., 2023) on top for mapping the pretrained dense embeddings $\boldsymbol{x} \in \mathbb{R}^d$ into a sparse embedding $\boldsymbol{z} \in \mathbb{R}^{d_z}$ with up to $k \ll d$ non-zero elements (i.e., $k$-sparse):

$$\boldsymbol{z} = \text{TopK}(\text{ReLU}(\boldsymbol{W}_{\text{enc}}(\boldsymbol{x} - \boldsymbol{b}_{pre}) + \boldsymbol{b}_{enc})), \tag{2}$$

$$\hat{\boldsymbol{x}} = \boldsymbol{W}_{\text{dec}}\boldsymbol{z} + \boldsymbol{b}_{\text{pre}}, \tag{3}$$

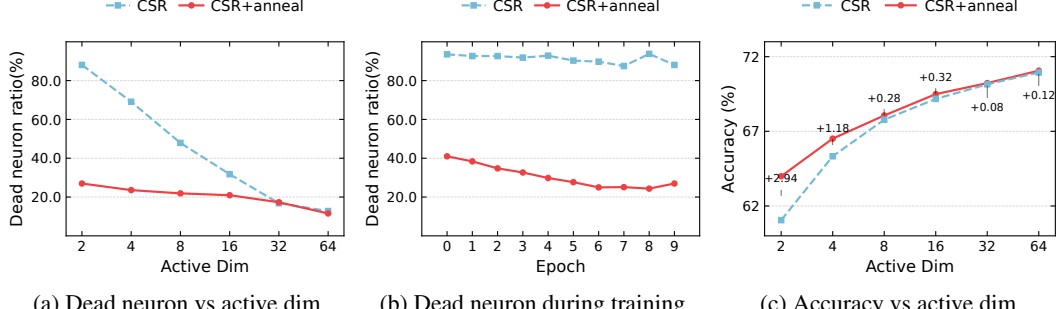

(a) Dead neuron vs active dim     (b) Dead neuron during training     (c) Accuracy vs active dim

Figure 2: **K-annealing analysis on ImageNet-1k with FF2048 as backbone**. (**Left**): Comparison of dead neuron ratio before and after applying k-annealing in different sparsity levels. (**Middle**): Dead neuron trend during training before and after applying annealing when $k = 2$. (**Right**): Evaluation results on ImageNet-1k with 1-NN accuracy as the main metric.

where the $\mathrm{TopK}$ operator keeps the top $k$ largest values while setting the others to zero, $\mathrm{ReLU}(x) = \max(x, 0)$ keeps non-negative elements, and $\boldsymbol{W}_{\mathrm{enc}}$ and $\boldsymbol{W}_{\mathrm{dec}}$ are the encoder and decoder matrices. The CSR model is jointly optimized via TopK sparse autoencoder (SAE) (Gao et al., 2024) and sparse contrastive learning (NCL) (Wang et al., 2024b). The overall training objective is,

$$\mathcal{L}_{\mathrm{CSR}} = \mathcal{L}(k) + \mathcal{L}(4k)/8 + \beta \mathcal{L}_{\mathrm{aux}} + \gamma \mathcal{L}_{\mathrm{SpCL}}. \tag{4}$$

The MSE loss $\mathcal{L}(k) = \|\boldsymbol{x} - \hat{\boldsymbol{x}}\|_2^2$ calculates the difference between original dense feature $\boldsymbol{x} \in \mathbb{R}^d$ and reconstructed dense feature $\hat{\boldsymbol{x}} \in \mathbb{R}^d$ from $k$-sparse embedding $\boldsymbol{z}$. Training with the multi-TopK loss $\mathcal{L}(k) + \mathcal{L}(4k)/8$ ensures that CSR could generalize to different $k$s at test time. The sparse contrastive loss $\mathcal{L}_{\mathrm{SpCL}}$ computes InfoNCE loss over sparse embeddings $\boldsymbol{z}$ as Wang et al. (2024b):

$$\mathcal{L}_{\mathrm{SpCL}} = -\frac{1}{|\mathcal{B}|} \sum_{i=1}^{|\mathcal{B}|} \log \frac{\exp \boldsymbol{z}_i^T \boldsymbol{z}_i}{\exp \boldsymbol{z}_i^T \boldsymbol{z}_i + \sum_{j \neq i}^{\mathcal{B}} \exp \boldsymbol{z}_i^T \boldsymbol{z}_j}. \tag{5}$$

Lastly, the auxiliary loss $\mathcal{L}_{\mathrm{aux}} = \|\boldsymbol{e} - \hat{\boldsymbol{e}}\|_2^2$ calculates the difference between the reconstruction error $\boldsymbol{e} = \boldsymbol{x} - \hat{\boldsymbol{x}}$ and the reconstruction using the top-$k_{\mathrm{aux}}$ dead latents $\hat{\boldsymbol{e}} = \boldsymbol{W}_{\mathrm{dec}}\boldsymbol{z}$, which is proposed by Gao et al. (2024) for reducing dead neuron's effect on performance degradation.

**Computational Complexity.** By exploiting short and sparse embeddings, both CSR and MRL significantly improve the memory and computational efficiency of embedding models. In particular, retrieval with a $k$-dimensional short embedding $\boldsymbol{z}$ requires only $\mathcal{O}(k)$ memory and compute to evaluate query–document similarity (instead of $\mathcal{O}(d)$ with $\boldsymbol{x}$). Likewise, storing a $k$-sparse embedding in compressed formats (e.g., CSR or CSC) incurs $\mathcal{O}(k)$ memory and enables $\mathcal{O}(k)$ compute via sparse matrix multiplication, which is natively supported in modern CPU/GPU libraries such as PyTorch. Wen et al. (2025) further show that CSR and MRL achieve comparable retrieval time at the same $k$. Hence, $k$ serves as a convenient surrogate for both memory and computational cost.

## 3  CSRv2: Tackling New Challenges under Ultra-sparsity

Although CSR achieves impressive performance by closely matching the accuracy of full-size embeddings at relatively high sparsity levels ($k = 8, 16, 32$), we observe that its performance deteriorates rapidly at extremely small values of $k$ (e.g., $k = 2, 4$). We refer to this regime as **ultra-sparsity**. In this section, we uncover several key reasons underlying CSR's failure in the ultra-sparse regime and show that it is actually largely fixable with several improved training techniques introduced here.

### 3.1  Tackling Massive Dead Neurons with k-Annealing

**The Massive Dead Neuron Phenomenon.** As discussed in Wen et al. (2025), a critical advantage of CSR over MRL is that sparse embeddings $\boldsymbol{z} \in \mathbb{R}^{d_z}$ can exploit a large number of hidden neurons $d_z \gg k$ for better feature expressivity, while only activating a few ($k$) for retrieval efficiency.

However, we observe that as $k \to 1$, dead neurons arise as a more severe problem. A dead neuron is a feature dimension that remains inactive on any data sample, indicating that it fails to represent anything useful. As shown in Figure 2a, the dead neuron ratio quickly increases as $k$ decreases, rising to 70% at $k = 4$ and reaching 90% at $k = 2$. This means that these ultra-sparse embeddings can only utilize 10% to 30% hidden dimensions, which greatly limits their representation power.

**Why Dead Neurons are more Severe under Ultra-sparsity.** Although CSR already integrates common remedies for dead neurons, such as auxiliary losses and multi-TopK strategies (Jermyn & Templeton, 2024; Gao et al., 2024), our experiments reveal that these approaches, effective at moderate sparsity ($k = 32, 64$), become largely ineffective when $k$ is extremely small. The difficulty is intrinsic: only the $k$ selected dimensions in each sparse code receive non-zero gradients, leaving the majority of neurons untrained. Under ultra-sparsity with only a handful of active dimensions, this issue becomes particularly severe. Moreover, once a neuron falls inactive, it receives no gradient signal and thus cannot recover, creating a self-reinforcing loop that further increases dead neurons.

**Alleviating dead neurons with $k$-annealing.** To alleviate this problem, we instead adopt a *curriculum learning* approach: we warm up the training with a sufficiently large initial sparsity level $k_{\text{init}}$ (by default $k_{\text{init}} = 64$), which avoids severe neuron inactivity and allows the model to learn a meaningful latent space in the early stage. As training proceeds, $k$ is gradually annealed toward the target ultra-sparsity $k_{\text{final}}$ (e.g., $k_{\text{final}} = 2$) using a linear schedule. Specifically, at epoch $t$ we set

$$k_t = (1 - p_t)\, k_{\text{init}} + p_t\, k_{\text{final}}, \qquad p_t = t/T, \qquad (6)$$

where $T$ is the total number of annealing steps. In practice, we perform annealing for 70% of training, after which $k$ is fixed at $k_{\text{final}}$. Analogous to simulated annealing, starting with a larger $k_{\text{init}}$ promotes exploration and diverse neuron activations, while the gradual annealing $k_{\text{init}} \to k_{\text{final}}$ sharpens the representations and enables stable convergence in the ultra-sparse regime.

We find this approach effectively maintains a low dead-neuron rate during training. As shown in Figure 2b, although dead neurons rise slightly at target sparsity, their final proportion is far lower than training directly with $k_{\text{final}}$. This indicates that a curriculum schedule provides richer gradients and avoids collapse into the dead-neuron regime. Similar to simulated annealing, a larger $k_{\text{init}}$ promotes exploration and diverse activations, while annealing gradually sharpens embeddings toward the ultra-sparse regime. Figure 2c confirms this, as k-annealing yields consistent performance gains across sparsity levels.

**Remark.** It is worth noting that LlamaScope (He et al., 2024) also employs a $k$-annealing strategy, but with a very different motivation and scope. Their annealing is applied only during the first 10% of training, reducing $k$ from the full embedding dimension ($k_{\text{init}} = d$) to a moderate sparsity level ($k_{\text{final}} = 50$) to accelerate convergence. In contrast, our method anneals $k$ over most of the training process, specifically to mitigate the massive dead neuron problem that arises under ultra-sparsity. Moreover, LlamaScope restricts annealing to SAEs, while we apply it to efficient embeddings. Thus, our finding that progressive $k$-annealing is critical for overcoming dead neurons at ultra-sparse embeddings still constitutes a novel and valuable contribution to the literature.

## 3.2 LEARNING DOWNSTREAM-ALIGNED FEATURES FROM NATURAL SUPERVISION

For ultra-sparse embeddings that activate only a few dimensions, the model must prioritize informative features and suppress noise. CSR, relying on self-supervised objectives like autoencoding and contrastive learning (Section 2), may be suboptimal. Its augmentation-based positives (e.g., cropping), though effective, transfer poorly when downstream tasks need properties ignored during training. (Ericsson et al., 2021). This weakness is exacerbated under ultra-sparsity, where noisy features are easily activated while informative ones are lost.

**Remedy: Sparse Supervised Contrastive Learning.** To bridge this gap, we follow the setting of MRL and adopt natural supervision, which is readily available in many retrieval tasks, to construct more accurate positive pairs. For example, in labeled datasets such as ImageNet, two random images from the same class can be used as a positive pair. In text retrieval datasets, query–document pairs naturally serve as positives. This supervision enables ultra-sparse embeddings to dedicate their limited active dimensions to encoding informative features that align with downstream applications, rather than wasting capacity on noisy features. Concretely, we replace CSR's self-supervised contrastive loss with a supervised contrastive loss (Khosla et al., 2020) applied to the $k$-sparse embed-

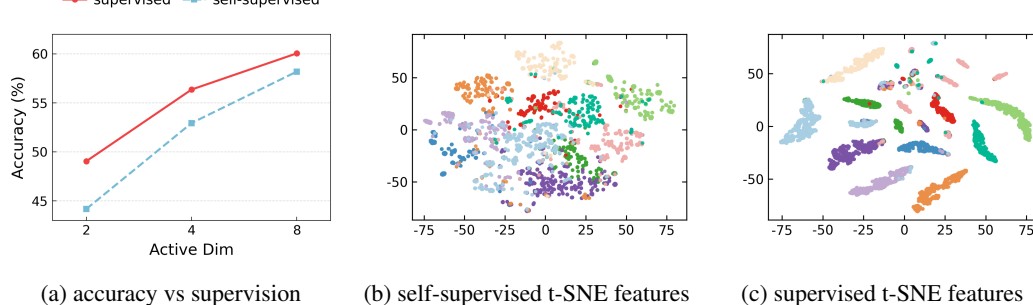

(a) accuracy vs supervision     (b) self-supervised t-SNE features     (c) supervised t-SNE features

Figure 3: **(Left)**: Supervision leads to performance increase in ultra-sparse setting with e5-Mistral-7B as backbone. **(Middle & Right)**: t-SNE visualization comparison on MTOPDomain (Li et al., 2020) before/after adding supervision when $k = 2$.

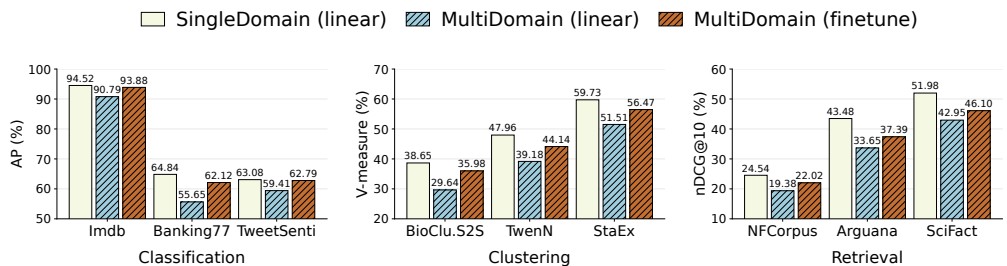

Figure 4: Comparison of CSRv2-linear trained on single-domain dataset, CSRv2-linear trained on multi-domain dataset and CSRv2 trained on multi-domain dataset in different tasks. e5-Mistral-7B is selected as backbone and training splits of all tasks in the same task type are combined for multi-domain.

dings:

$$\mathcal{L}_{\text{SpSCL}}(k) = -\frac{1}{|\mathcal{B}|} \sum_{i=1}^{|\mathcal{B}|} \log \frac{\sum_{p \in \mathcal{P}(i)} e^{\boldsymbol{z}_i^T \boldsymbol{z}_p}}{\sum_{p \in \mathcal{P}(i)} e^{\boldsymbol{z}_i^T \boldsymbol{z}_p} + \sum_{n \in \mathcal{N}(i)} e^{\boldsymbol{z}_i^T \boldsymbol{z}_n}}, \tag{7}$$

where $\mathcal{P}(i)$ and $\mathcal{N}(i)$ denote the sets of positive and negative samples derived from natural supervision. Specifically, for classification and clustering tasks, samples with the same label are treated as positives, while others are negatives. For retrieval and reranking tasks, each query and its corresponding documents are positives. For semantic textual similarity, sentence pairs with a similarity score above 3 are positives. For pair classification, sentence pairs with label 1, indicating strong correlation, are positives. A detailed description of these tasks is provided in Appendix B.

From Figure 3, we observe that supervised training yields clear performance gain in ultra-sparse settings. Moreover, supervision produces sparse features that are far more discriminative across classes. It demonstrates that these supervision signals provide clearer signals for training ultra-sparse embeddings. More detailed ablation on applying natural supervision to CSR is available in Section 4.3. The community has so far curated abundant pretraining-scale paired text data for training retrieval models, such as 65M Q-A pairs (Lewis et al., 2021). Therefore, it would be quite useful to be able to leverage large-scale supervision.

## 3.3 MITIGATING MULTI-DOMAIN TRAINING GAPS VIA FINETUNING

A notable property of CSR is that it can outperform MRL (which requires full finetuning) by training only a simple encoder with a single linear layer. However, this design also limits CSR's ability to fully exploit the potential of sparse embeddings, particularly when deploying a single model across multiple downstream tasks. As shown in Figure 4, CSR with only a linear layer experiences clear performance drop under joint training with multi-domain data in different task types, reflecting the limited capacity of such a shallow adaptation.

Table 2: **Performance and retrieval efficiency on six text embedding tasks with e5-Mistral-7B**. Since e5 does not natively support MRL or CSR, we enable a fair comparison by training all methods on the same backbone, data, and configurations. For retrieval efficiency, experiments are conducted with a 1M database, and results are reported as retrieval time relative to CSRv2 at $k = 2$.

| Active Dim | Method | Retrieval Time | Classifi. ACC ↑ | Clust. V-measure ↑ | Retrieval nDCG@10 ↑ | STS Spearman ↑ | PairClassifi. AP ↑ | Rerank. MAP ↑ | Avg. |
|---|---|---|---|---|---|---|---|---|---|
| 4096 | e5-Mistral-7B | 306.46× | 80.67 | 51.55 | 49.35 | 84.11 | 91.77 | 69.52 | 69.99 |
| 4096 | MRL | 301.86× | 80.46 | 50.94 | 48.75 | 83.78 | 90.44 | 68.86 | 69.49 |
| | CSR | 197.52× | 80.54 | 51.13 | 49.13 | 83.94 | 90.99 | 68.96 | 69.70 |
| | CSRv2-linear | 196.04× | 80.55 | 51.19 | 49.07 | 84.02 | 91.48 | 69.02 | 69.76 |
| | CSRv2 | 201.42× | 80.49 | 51.34 | 49.16 | 83.94 | 91.70 | 69.18 | 69.80 |
| 64 | MRL | 17.30× | 66.58 | 47.76 | 44.11 | 77.46 | 78.46 | 62.72 | 61.86 |
| | CSR | 14.92× | 79.50 | 48.36 | 45.22 | 82.10 | 87.29 | 64.86 | 66.68 |
| | CSRv2-linear | 14.53× | 80.29 | 48.35 | 47.92 | 82.09 | 88.55 | 66.54 | 67.58 |
| | CSRv2 | 14.17× | 79.98 | 49.53 | 47.92 | 82.90 | 90.46 | 67.34 | 68.08 |
| 16 | MRL | 7.77× | 54.64 | 42.03 | 34.33 | 68.18 | 59.22 | 56.16 | 51.93 |
| | CSR | 3.53× | 75.61 | 45.12 | 34.79 | 77.30 | 84.28 | 59.86 | 62.83 |
| | CSRv2-linear | 3.51× | 77.08 | 46.58 | 39.60 | 79.37 | 85.38 | 62.31 | 64.26 |
| | CSRv2 | 3.51× | 77.79 | 47.97 | 43.38 | 79.94 | 86.50 | 64.36 | 65.76 |
| 4 | MRL | 6.29× | 43.84 | 33.14 | 24.55 | 56.51 | 37.36 | 44.72 | 40.83 |
| | CSR | 1.62× | 67.22 | 39.25 | 23.54 | 70.13 | 74.44 | 48.57 | 52.94 |
| | CSRv2-linear | 1.65× | 73.55 | 42.96 | 34.31 | 73.31 | 74.17 | 56.08 | 58.62 |
| | CSRv2 | 1.63× | 74.26 | 43.85 | 39.04 | 75.69 | 74.90 | 62.93 | 61.01 |
| 2 | MRL | 6.20× | 34.84 | 26.13 | 16.63 | 52.14 | 26.67 | 40.30 | 33.81 |
| | CSR | 1.01× | 52.50 | 35.20 | 16.14 | 62.93 | 52.95 | 46.77 | 44.33 |
| | CSRv2-linear | 1.01× | 66.43 | 39.07 | 31.58 | 67.91 | 57.39 | 53.32 | 53.35 |
| | CSRv2 | 1.00× | 71.59 | 41.29 | 37.48 | 73.82 | 62.46 | 60.91 | 58.38 |

To fully unlock the potential of sparse embeddings and push CSR to its limit, we adopt the same setting as MRL: applying the $\mathrm{TopK}$ operator to the output embeddings of the backbone network and finetuning the entire model. Figure 4 shows that full finetuning effectively mitigates the performance drop observed under the linear setting, recovering performance comparable to domain-specific CSR training (additional details are provided in Appendix C).

Building on all these findings above, we propose the following improved sparse training objective:

$$\mathcal{L}_{\mathrm{CSRv2}} = \mathcal{L}(k_t) + \tfrac{1}{8}\mathcal{L}(4k_t) + \beta\mathcal{L}_{\mathrm{aux}} + \gamma\mathcal{L}_{\mathrm{SpSCL}}(k_t), \tag{8}$$

where $k_t$ is the annealed sparsity level at step $t$ (Eq. 6) and $\mathcal{L}_{\mathrm{SpSCL}}$ denotes the sparse supervised contrastive loss (Eq. 7).

We designate the fully finetuned model as **CSRv2**, and the variant that finetunes only a linear layer on top (as in CSR) as **CSRv2-linear**. TopK SAE (Gao et al., 2024) finds that using $L(k) + \frac{L(4k)}{8}$ is enough to obtain progressive representation over all $k$. We find similar phenomena for CSR and thus follow this common practice. The improved training recipe remains as simple and generic as the original CSR without introducing more training objectives. In the experiments that follow, we are able to show that both CSRv2 and CSRv2-linear deliver significant gains over CSR and MRL, particularly in the ultra-sparse regime. Furthermore, the fully finetuned CSRv2 sets a new performance–efficiency frontier for adaptive embeddings, surpassing MRL by up to 25% in absolute accuracy under the same setting. During inference, the embedding produced by the backbone first goes through an encoder which projects it onto a high dimensional vector (e.g. 16384). Afterwards, the TopK values in the vector are kept while others are set 0, with no normalization applied.

## 4 EXPERIMENTS

In this section, we comprehensively evaluate the effectiveness of CSRv2. For language representation, we evaluate on tasks in Appendix B. For visual representation, we conduct experiments on ImageNet-1k (Deng et al., 2009) and evaluate using 1-NN accuracy (Kusupati et al., 2022). More-

Table 3: **Performance comparison with Qwen3-Embedding-4B** (Zhang et al., 2025b), a state-of-the-art embedding model on MTEB that natively supports MRL. Backbone results are shown in the first line and first/second largest value on each active dimension is **bold** / underlined.

| Active Dim | Method | Classifi. ACC ↑ | Clust. V-measure ↑ | Retrieval nDCG@10 ↑ | STS Spearman ↑ | PairClassifi. AP ↑ | Rerank. MAP ↑ | Avg. |
|---|---|---|---|---|---|---|---|---|
| 2560 | Qwen3-Embed-4B | 85.79 | 55.27 | 58.37 | 88.63 | 91.42 | 72.03 | 74.92 |
| 2560 | MRL | 85.38 | 55.04 | 58.31 | 88.02 | 91.27 | 71.64 | 74.58 |
| | CSR | 85.49 | 54.83 | 58.21 | 88.64 | 91.23 | 71.84 | 74.70 |
| | CSRv2-linear | 85.32 | 55.43 | **58.74** | **89.05** | 91.03 | **72.25** | **74.99** |
| | CSRv2 | **85.58** | **55.91** | 58.23 | 88.47 | **91.39** | 71.98 | 74.91 |
| 64 | MRL | 83.42 | 53.73 | 44.13 | **86.60** | 88.08 | 69.61 | 70.54 |
| | CSR | 83.94 | 52.36 | 51.51 | 85.33 | 90.54 | 70.11 | 71.10 |
| | CSRv2-linear | 84.03 | 53.19 | 53.22 | 85.88 | 90.72 | 71.13 | 72.31 |
| | CSRv2 | **84.28** | **54.57** | **55.64** | 86.32 | **90.90** | **71.64** | **72.79** |
| 16 | MRL | 75.22 | 47.24 | 20.40 | 79.21 | 73.29 | 60.82 | 58.89 |
| | CSR | 78.60 | 49.08 | 35.66 | 82.08 | 85.80 | 65.00 | 64.66 |
| | CSRv2-linear | 80.71 | 51.48 | 39.09 | 82.15 | 88.94 | 67.64 | 67.20 |
| | CSRv2 | **82.03** | **53.86** | **45.09** | **82.63** | **90.42** | **69.89** | **68.98** |
| 4 | MRL | 48.27 | 32.08 | 6.59 | 53.11 | 30.73 | 40.59 | 36.74 |
| | CSR | 57.39 | 36.03 | 16.27 | 64.13 | 64.29 | 50.26 | 46.66 |
| | CSRv2-linear | 71.59 | 43.24 | 24.82 | 72.11 | 77.74 | 56.94 | 56.76 |
| | CSRv2 | **80.20** | **48.27** | **29.71** | **77.94** | **82.28** | **62.98** | **62.41** |
| 2 | MRL | 26.47 | 24.20 | 5.23 | 30.22 | 18.46 | 32.43 | 22.84 |
| | CSR | 41.83 | 30.02 | 9.37 | 51.27 | 54.60 | 44.20 | 36.29 |
| | CSRv2-linear | 66.95 | 39.22 | 18.47 | 71.56 | **77.95** | 54.67 | 53.41 |
| | CSRv2 | **76.22** | **46.02** | **23.93** | **74.88** | 75.24 | **59.52** | **58.53** |

over, we conduct efficiency analysis and empirical analysis on ablation of each component and dead neurons. Case study of representation interpretability for a more detailed assessment of the advantages and potential of CSRv2 is proposed in Appendix F.

## 4.1 BENCHMARK PERFORMANCE

**Evaluation under controlled setup.** For fair comparison, we adopt e5-Mistral-7B (Wang et al., 2023) as backbone and finetune it on MTEB datasets to ensure MRL aligns with CSRv2 domains. Table 2 reports task-type-specific results on six task types commonly adopted in past works (Zhang et al., 2024) (Li et al., 2024a) (Lee et al., 2024a) in MTEB (Muennighoff et al., 2022), where CSRv2 is trained on all train splits of the same task type. Under equal activation dimensions, CSRv2 consistently outperforms CSR, with up to 14% gains in the ultra-sparse case $k = 2$. Notably, CSRv2 also surpasses MRL: at $k = 2$, it exceeds MRL's dense 16-dim embedding and even outperforms 64-dim dense embeddings in text classification. Efficiency tests on a 1M database further show CSRv2's ultra-sparse embeddings achieve a $300\times$ retrieval speedup over the backbone and $7\times$ faster retrieval than MRL's dense embeddings of similar accuracy. More detailed results and implementation details appear in Appendix C.

**Evaluation on State-of-the-art Qwen3 Models.** We further evaluate on Qwen3-Embedding-4B, whose series leads the MTEB leaderboard, with even the 0.6B model surpassing prior 7B results. Unlike E5-Mistral-7B, Qwen3 integrates MRL into training, producing embeddings naturally aligned with it. As shown in Table 3, CSRv2 consistently outperforms both MRL and CSR at equal compression. In cross-level comparisons, CSRv2 at $k = 16$ rivals MRL at $k = 64$, and CSRv2 at $k = 2$ rivals MRL at $k = 16$, highlighting its adaptability across backbones and sparsity levels.

**Evaluation Comparison with SPLADE Sparse Retrieval Model.** As a representative SOTA for learning-based sparse retrieval, SPLADE (Lassance et al., 2024) generally achieves high performance with $\sim 3\%$ activation. However, our MTEB comparison reveals its fragility in high-sparsity settings. SPLADEv3 at $K = 16$ clearly lags behind CSRv2, and remarkably, its retrieval quality at $K = 8$ trails CSRv2 at $K = 2$, highlighting our method's superior compression efficiency. (See Appendix C.5).

**Zero-shot Evaluation in GraphRAG System.** We further assess CSRv2's zero-shot capability on medical and novel domains using the GraphRAG Benchmark (Xiang et al., 2025), focusing on retrieval accuracy and generation quality. Despite being trained solely on MTEB (zero-shot setting), CSRv2 shows significantly less degradation than MRL, indicating superior robustness to unseen data distributions. Detailed results and setup are provided in Appendix E.

**Visual Embedding Evaluation on ImageNet-1k.** Figure 5a demonstrates CSRv2's performance on ImageNet-1k with pre-trained ResNet-50 noted as FF2048 in the MRL (Kusupati et al., 2022) as backbone. We find that CSRv2 achieves continuous improvement in classification performance compared to CSR and MRL. This phenomenon is particularly prominent in the extremely sparse case, where CSRv2 achieves a 6% 1-NN accuracy increase over CSR and 20% over MRL. More detailed results and experiment setup are in Appendix D.

## 4.2 EFFICIENCY ANALYSIS

In Figure 5c, we evaluate CSRv2 and MRL retrieval efficiency under hidden dimension $\mathbb{R}^h$ and active dimension $K$ on a 1M-scale database. Retrieval time grows roughly linearly with $d$ as predicted by $O(dk)$, though GPU architecture also influences performance. In ultra-sparse cases ($k = 2$), CSRv2 leverages GPU sparse accelerators (e.g., Sparse Tensor Core, cuSPARSE) to run over $6\times$ faster than MRL. As sparsity decreases ($k = 32$), dense-optimized libraries (e.g., cuBLAS) reduce dense operators' overhead, shrinking CSRv2's advantage to $2.2\times$. Thus, CSRv2 excels in extreme sparsity while maintaining stable gains in general sparse settings. Experiment setups and more discussions on encoding, indexing and retrieval are presented in Appendix G.1.

## 4.3 EMPIRICAL ANALYSIS

**Ablation.** Table 4 reports ablations of CSRv2 components. Supervision proves most effective for compression, while anneal alone yields little gain. Yet combining them (CSRv2-linear) outperforms adding supervision alone, showing synergy: anneal promotes feature orthogonality and subspace expansion, while supervision directs semantic learning, where the two play different but complementary roles. Finetuning further aligns backbone embeddings with sparse objectives, adding 5% improvement at $k = 2$.

**Dead Neurons.** Figure 5b shows dead neuron fractions across components. While adding unsupervised contrastive loss in CSR yields more independent features and fewer dead neurons in sparse embedding (e.g. $k = 32$), CSR still suffers severe dead neuron issues in ultra-sparse cases (e.g. $k = 2$). Anneal distributes semantic features into a broader hidden subspace, reducing dead neurons by 70% at $k = 2$. Natural supervision further lowers them to about 20%. Finetuning brings little improvement, likely because the TopK strategy only aligns backbone embeddings with sparse objectives rather than fostering orthogonal representations.

**K-Schedule Sensitivity Analysis.** We test on k-annealing's sensitivity on three perspectives: k-schedule shape, length (i.e. ratio of steps before $k$ reaches target sparsity level) and $k$'s initialization. Results show that different k-schedule results in relatively stable increase in performance improvement, while our selected setting: initialized to 64, annealing to target sparsity level at 70% step, and linear-annealing strategy achieves the best performance. More detailed results are in Appendix G.2.

**Further Discussions.** Moreover, we have conducted several experiments, which provide potential directions for future exploration. These discussions are analysis on unbalanced weightable settings for MRL and CSRv2 finetuning (Appendix G.3), emergence of superclass separability in sparsity representation (Appendix H.1), MRL-SAE exploration (Appendix H.5) and quantized comparison at fixed memory cost (Appendix H.2). Furthermore, CSRv2 can be potentially applied in vector quantization due to its sparse structure, with a brief discussion in Appendix H.3.

## 5 CONCLUDING REMARKS

Unlike prior methods (CSR, MRL) that fail once $k \leq 4$, CSRv2 provides the first principled recipe that makes ultra-sparsity viable. The central insight is that **ultra-sparsity is not merely a parameter regime but a qualitatively different optimization problem**: (i) standard self-supervised losses misalign with downstream semantics when only two or four features remain, and (ii) dead neurons

Table 4: **Performance Ablation Comparison**: We perform ablation study with e5-Mistral-7B as backbone through task-type-specific evaluation and average performance of all task types is presented. We mark improvement of different combinations relative to CSR with green, while performance gap between MRL and CSR with red.

| | Components | | | Active Dimension | | | | | |
|---|---|---|---|---|---|---|---|---|---|
| | anneal | supervise | finetune | 64 | 32 | 16 | 8 | 4 | 2 |
| MRL | - | - | ✓ | 61.47 (-5.21) | 56.37 (-8.23) | 51.85 (-9.53) | 47.09 (-11.10) | 40.83 (-12.11) | 33.81 (-10.36) |
| CSR | ✗ | ✗ | ✗ | 66.68 | 64.60 | 61.38 | 58.19 | 52.94 | 44.17 |
| + anneal | ✓ | ✗ | ✗ | 67.35 (+0.67) | 65.24 (+0.64) | 61.91 (+0.53) | 58.79 (+0.60) | 54.55 (+1.61) | 45.33 (+1.16) |
| + supervise | ✗ | ✓ | ✗ | 67.32 (+0.64) | 65.54 (+0.94) | 62.95 (+1.57) | 60.05 (+1.86) | 56.36 (+3.42) | 49.05 (+4.88) |
| CSRv2-linear | ✓ | ✓ | ✗ | 67.58 (+0.90) | 65.83 (+1.23) | 63.73 (+2.35) | 61.53 (+3.34) | 58.62 (+5.68) | 53.25 (+9.08) |
| CSRv2 | ✓ | ✓ | ✓ | **68.08** (+1.40) | **66.70** (+2.10) | **65.22** (+3.84) | **63.76** (+5.57) | **61.01** (+8.07) | **58.34** (+14.17) |

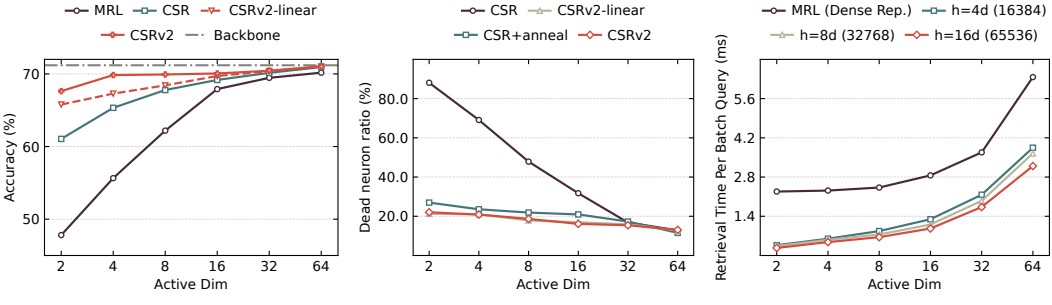

(a) Visual embedding evaluation     (b) Dead neuron comparison     (c) Retrieval time per active dim

Figure 5: **(Left)**: Visual representation results on ImageNet-1k with FF2048 as backbone. "Backbone" serves as the evaluation results on FF2048 without finetuning for consistent reference. **(Middle)**: Dead neuron trend with different components under varying compression levels. **(Right)**: Efficiency analysis in 1M database size with e5-Mistral-7B as backbone.

accumulate irreversibly without curriculum. CSRv2 introduces two non-trivial modifications motivated by this diagnosis: a progressive $k$-annealing schedule that preserves gradient flow across neurons until late training, and a supervised sparse contrastive objective that reallocates the few active features to carry semantic signal. These mechanisms are essential for surviving the ultra-sparse regime and go beyond "better tuning" of CSR's original objective.

A key open challenge is the $k = 1$ regime, where CSRv2 still suffers from severe dead neurons and sharp degradation (Appendix H.4). Since $k = 1$ effectively reduces to clustering (mapping each input to a one-shot label), future work could explore clustering-inspired approaches, such as prototype or vector quantization, balanced assignment, entropy regularization, or optimal transport. Extending CSRv2 into this extreme setting remains an important direction, while the practically useful ultra-sparse range $k \in \{2, 4, 8\}$ already offers substantial efficiency gains with competitive accuracy.

**Ethics Statement.** This work adheres to the ICLR Code of Ethics. Although our proposed methods are broadly applicable, their deployment in real-world scenarios may carry societal considerations, particularly regarding bias, fairness, and privacy. We advocate for responsible application of our techniques and disclose that we have no conflicts of interest.

**Reproducibility.** We provide comprehensive details of our methodology, datasets, model architectures, and evaluation protocols in both the main text and Appendix. Full mathematical derivations and additional experimental results are included in the Appendix. We publicly release the source code and scripts to facilitate complete reproduction of our experiments in https://github.com/Y-Research-SBU/CSRv2.

ACKNOWLEDGEMENTS

Yifei Wang and Stefanie Jegelka were supported in part by the NSF AI Institute TILOS (NSF CCF-2112665), and an Alexander von Humboldt Professorship.

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

## A    ADDITIONAL RELATED WORK

**LLM-based Text Embeddings.**    The integration of Large Language Model into text embedding generation has been a hot topic due to LLM's extraordinary capability of comprehensive semantic understanding. This integration has led to many embedding models that have demonstrated excellent performance in multiple domains, multiple tasks, and multiple languages, such as GritLM (Muennighoff et al., 2024), e5-Mistral-7B-instruct (Wang et al., 2023), Gemini Embedding (Lee et al., 2025), Qwen3 Embedding (Zhang et al., 2025b) and Linq-Embed-Mistral (Choi et al., 2024).

Generally, the techniques utilized in training these models can mainly be classified into two categories. One approach is utilizing LLMs for text augmentation or synthetic data generation, therefore expanding the domain covered by model training. Jina-v3 (Sturua et al., 2024), Gecko (Lee et al., 2024b) and Tart (Lu et al., 2024) utilize LLM to generate synthetic examples to enhance task-wise generation and expand targeted failure cases. NV-Embed-v2 (Moreira et al., 2024), E5-Mistral (Cheng et al., 2023) and SWIM-X (Thakur et al., 2023) employ LLM to provide higher-quality supervision signals for existing embedding training.

Another approach is directly adapting LLMs themselves to serve as text embedding models, therefore transfer knowledge from large LLMs to more efficient embedding models. Generally, this approach takes one pretrained LLM as backbone such as Mistral 7B (Wang et al., 2023) and finetune with parameter-efficient finetuning strategies including BitFit (Zaken et al., 2021) and LoRA (Hu et al., 2022). Current innovation for LLM adaptation to embedding generation focus on three aspects: design of positive/negative pairs, multi-stage learning and instruction tuning. For design of positive/negative pairs, (Gao et al., 2021) proposes an unsupervised contrastive learning framework for advancing sentence embeddings, where augmented unlabeled sentences are seen as positive pairs. NV-Retriever (Moreira et al., 2024) filters out potential false negatives by comparing candidate negatives against the positive relevance score. Granite Embedding models (Awasthy et al., 2025) use additional bidirectional signal to expand negatives in retrieval set. Similarly, You et al. (2024) propose a contrastive feature recalibration approach to mitigate spurious correlations and enhance group robustness without relying on group annotations. For multi-stage training, NV-Embed (Lee et al., 2024a) takes a two-stage contrastive instruction-tuning approach that first trains on retrieval datasets with in-batch and hard negatives, then blends in non-retrieval tasks without in-batch negatives, yielding strong improvements in both retrieval and general embedding tasks. Qwen3 Embedding model series (Zhang et al., 2025b) take a three-step pipeline that first performs large-scale weakly supervised pre-training on synthetic data, then finetunes with high-quality supervised and selected synthetic datasets, and finally applies model merging (Li et al., 2024b) to boost robustness and generalization. For instruction tuning, Inbedder (Peng et al., 2024) treats instructions as questions and derives embeddings from the expected answers rather than concatenating instructions with inputs. E5-Mistral (Cheng et al., 2023) employed an asymmetric instruction strategy that initially applies instructions only to the query side which has been proven efficient in retrieval tasks by numerous subsequent works.

**Adaptive Representations Learning for Text Embedding Compression.**    Early work for text embedding sparsity focuses on directly mapping text to sparse vectors or use token-wise late interaction, with some recent work carried out following this approach. The SPLADE series (Formal et al., 2021b) (Formal et al., 2021a) introduced a BERT-based model for learning sparse, interpretable text representations via explicit sparsity regularization and log-saturation, enabling efficient inverted index retrieval. PromptReps (Zhuang et al., 2024a) prompts LLM to generate a single-word representation of each text and sparsify the logits of that prediction by filtering to document tokens while applying ReLU and log-saturation. Mirzadeh et al. (2023) proposes a "relufication" sparsity strategy where non-ReLU activations in pretrained LLMs are replaced (and sometimes supplemented) with ReLU layers to induce high activation sparsity. Nguyen et al. (2024) uses probabilistic term expansion control to transform dense text embeddings in multimodal retrieval into sparse, vocabulary-aligned vectors while preserving effectiveness. Wang et al. (2024a) introduces Q-Sparse, a method that achieves full activation sparsity in large language models by applying TopK sparsification to linear projections and using the straight-through estimator for training Moreover, You et al. (2025) reveal that spurious memorization — where a small set of neurons overfit to non-causal patterns — can lead to biased representations and degraded generalization. Understanding and mitigating such effects provides complementary insight to sparsity-based embedding learning.

Matryoshka Representation Learning (Kusupati et al., 2022) (MRL) pioneers text embedding compression in recent years via training with truncated dimensions. Proposed in 2022, MRL demonstrates adaptive-length embeddings for large-scale retrieval and classification including NLP settings, leading to various works that focus on adapting MRL to embedding model settings. Li et al. (2024c) extends MRL by introducing a second scalability dimension, enabling embeddings to be truncated along both model layers and embedding sizes. Zhuang et al. (2024b) combines fixed-size sub-model finetuning with masked autoencoder pre-training, introduces a new structured training strategy for 2D Matryoshka embeddings. Yoon et al. (2024) transforms arbitrary embeddings generated by embedding models or APIs into embeddings with Matryoshka properties in both unsupervised and supervised setups. Various open-sourced embedding models, such as Jina-v3 (Sturua et al., 2024) and Qwen3-Embedding series (Zhang et al., 2025b); and commercial APIs, such as Gemini (Lee et al., 2025), have supported MRL dimension truncation.

Another promising direction for text embedding sparsification is Sparse Autoencoder, which grows from sparse coding/dictionary learning to tackle polysemanticity by disentangling features, are now scaled to frontier LLMs and widely used for mechanistic interpretability (Cunningham et al., 2023) (Yan et al., 2025). Rajamanoharan et al. (2024) introduces Gated SAE that solves the systematic underestimation of feature activations caused by L1 penalty and requires half as many firing features to achieve comparable reconstruction fidelity. Gao et al. (2024) utilizes k-sparse autoencoders as a replacement of traditional L1-based sparsity, preventing activation shrinkage, reducing dead latents, and yielding cleaner scaling laws with more interpretable and effective features. Lan et al. (2024) employs SAE to discover monosemantic features within language models, revealing a high degree of similarity and potential universality in these learned sparse feature spaces across diverse LLM architectures. Duan et al. (2024a;b) explore sparse representations via principled Bayesian gamma priors in deep generative models. Wen et al. (2025) leverages contrastive objectives for preserving semantic quality, achieving results close to those of backbone embeddings in the downstream tasks when only 32 neurons are activated.

**Orthogonal Efficiency Techniques.** Quantization and hashing compress embedding *values* rather than reducing active dimensions. Product quantization and its optimized variants approximate distances with compact codes (Jégou et al., 2011; Ge et al., 2013), while binary hashing methods such as Spectral Hashing and ITQ yield extremely small codes with Hamming-distance search (Weiss et al., 2008; Gong et al., 2011). Model-side low-bit quantization of Transformer encoders further reduces memory and latency (Shen et al., 2019). These techniques are orthogonal and can be combined with sparse embeddings (e.g., PQ over nonzero coordinates or low-bit storage of sparse values), jointly improving storage and retrieval throughput.

## B  TASKS

We cover 6 types of tasks in this paper: classification, clustering, retrieval, pair classification, semantic textual similarity and reranking. They are taken from MTEB (Muennighoff et al., 2022) and include the vast majority of the tasks in the MTEB English Leaderboard, as well as some multilingual tasks.

- **Classification**: Classification involves 10 tasks, which are divided into general tasks and specialized tasks. General tasks include AmazonMassiveDomain (FitzGerald et al., 2022), AmazonMassiveScenario (FitzGerald et al., 2022) , MTOPIntent (Li et al., 2020), and MTOPDomain (Li et al., 2020) for multilingual natural language understanding, IMDb (Maas et al., 2011), TweetSentimentExtraction (Maggie et al., 2020) and Emotion (Saravia et al., 2018) for sentiment analysis. Specialized tasks include AmazonCounterfactual (O'Neill et al., 2021) for counterfactual detection in product reviews, ToxicConversation50k (cjadams et al., 2019) for detection of toxic speech and prejudice, Banking77 Casanueva et al. (2020) for financial intent recognition.

- **Clustering**: Clustering involves 8 tasks. These tasks are BiorxivClusteringP2P, BiorxivClusteringS2S [1], MedrxivClusteringP2P, MedrxivClusteringS2S [2] and ArxivClusteringS2S

---

[1]https://api.biorxiv.org/

[2]https://api.medrxiv.org/

(University, 2025) for research field clustering, TwentyNewsGroups [3] for news topics identification, StackExchangeP2P and StackExchange (Geigle et al., 2021) for clustering of titles from 121 stackexchanges.

- **Retrieval**: Retrieval involves 8 tasks. These tasks are Arguana (Wachsmuth et al., 2018) and NFCorpus (Boteva et al., 2016) for medical information retrieval, CQADupstackGaming (Hoogeveen et al., 2015) and CQADupstackUnix (Hoogeveen et al., 2015) for web community retrieval, ClimateFEVERHardNegatives (Wadden et al., 2020a) for climate-change retrieval, FiQA2018 (Maia et al., 2018) for financial retrieval, SCIDOCS (Cohan et al., 2020b) and SciFact (Wadden et al., 2020b) for academic retrieval.

- **Semantic Textual Similarity (STS)**: STS includes 10 tasks. These tasks include general-domain semantic comprehension tasks STS12 (Agirre et al., 2012), STS13 (Agirre et al., 2013), STS14 (Bandhakavi et al., 2014), STS15 (Biçici, 2015), STS16 (Nakov et al., 2019), STSBenchmark (May, 2021), SICK-R (Marelli et al., 2014), STS17 (Cer et al., 2017) and STS22 (Chen et al., 2022) and medical domain semantic comprehension task BIOSSES (Soğancıoğlu et al., 2017).

- **Pair Classification**: Pair Classification includes two tasks, with SprintDuplicateQuestions (Shah et al., 2018) for programming domain and TwitterURLCorpus (Lan et al., 2017) for social media (Tweet) domain.

- **Reranking**: Reranking includes 3 tasks, which are AskUbuntuDupQuestions (Wang et al., 2021) and StackOverflowDupQuestions (Liu et al., 2018) for reranking of related programming blogs and SciDocsRR (Cohan et al., 2020a) for reranking of scientific papers.

## C    EXPERIMENT DETAILS ON TEXT REPRESENTATIONS

### C.1    EVALUATION METRICS

We adopt the standardized evaluation protocols established by the Massive Text Embedding Benchmark (MTEB) (Muennighoff et al., 2022). Specifically:

- For classification tasks, we train a logistic regression classifier on the embedded training split and report its accuracy on the test split.

- For clustering tasks, we apply mini-batch k-means to the embedded training data and evaluate performance on the test split using the V-measure.

- For retrieval tasks, we compute normalized Discounted Cumulative Gain at rank 10 (nDCG@10), where document-query relevance scores are derived from cosine similarity between embeddings.

- For semantic textual similarity (STS) tasks, we measure the Spearman rank correlation coefficient between the ground-truth similarity scores and the cosine similarities of the corresponding sentence embeddings.

- For pair classification tasks, we evaluate using cosine-similarity–based average precision, with decision thresholds determined by optimizing over similarity scores on the validation set.

- For reranking tasks, we report Mean Average Precision (MAP), again using cosine similarity as the scoring function.

To assess retrieval efficiency, we construct a unified query set by aggregating all queries from the aforementioned retrieval and reranking datasets, and a corresponding document database by merging their respective corpora. All efficiency metrics are computed over this consolidated benchmark setup.

### C.2    EXPERIMENT SETUP

We select e5-Mistral-7B (Wang et al., 2023) and Qwen3-Embedding-4B (Zhang et al., 2025b) as our backbone embedding models and evaluate their performance across six task categories defined

---

[3]https://scikit-learn.org/0.19/datasets/twenty_newsgroups.html

in the Massive Text Embedding Benchmark (MTEB) (Muennighoff et al., 2022). For each task category, we restrict our evaluation to English-language datasets and the English subsets of multilingual datasets included in the MTEB leaderboard[4] . This yields a total of 10 classification (**Classif.**), 8 clustering (**Clust.**), 8 retrieval (**Retrieval**), 10 semantic textual similarity (**STS**), 2 pair classification (**PairClassifi.**), and 3 reranking (**Rerank.**) datasets in our experimental suite.

We adopt a *task-type-specific* evaluation pipeline: for each task type, we aggregate the training splits of all constituent datasets to form a unified training set, while preserving the original test split of each individual dataset for performance evaluation. This pipeline is applied consistently across all six task types using the aforementioned datasets.

All experiments are conducted on a server equipped with 8 NVIDIA A100-SXM4-40GB GPUs, except for backbone finetuning, which is performed on a separate server with 8 H20-NVLink GPUs (96 GB memory each).

### C.3 IMPLEMENTATION DETAILS

To ensure a fair comparison between MRL and CSRv2 on the MTEB benchmark, particularly with respect to domain alignment, we select a backbone model that is not natively supported by MRL: e5-Mistral-7B (Wang et al., 2023). We then finetune this model on a carefully curated collection of multi-domain datasets. Specifically, the training data is drawn from three complementary sources: (i) datasets included in the MTEB benchmark (Muennighoff et al., 2022), (ii) the embedding training data collection curated by the Sentence Transformers team[5] , and (iii) a suite of public retrieval datasets introduced in Zhang et al. (2025a). During preprocessing, we first deduplicate datasets that appear across multiple collections. Subsequently, following the natural supervision strategy outlined in Section 3.2, we sample up to 20,000 sentence pairs per dataset, resulting in a consolidated training corpus of approximately one million examples.

We finetune e5-Mistral-7B on this corpus using a batch size of 2048, which is a scale commonly adopted by existing MRL-compatible models. Full details of the hyperparameter configuration are provided in Table 5. In contrast, the Qwen3-Embedding-4B model (Zhang et al., 2025b) already incorporates native MRL support; thus, no additional finetuning is required for this backbone.

Table 5: Implementation details on MRL finetuning.

| Backbone | Batch Size | LoRA $r$ | LoRA $\alpha$ | lr | epoch | warmup | weight decay | MRL dim | MRL $c_m$ |
|---|---|---|---|---|---|---|---|---|---|
| e5-Mistral-7B | 2048 | 8 | 16 | 2e-5 | 10 | 1000 | 0.1 | 1,2,4...,4096 | $\{1, 1, \ldots, 1\}$ |

For backbone finetuning, we adopt a methodology closely aligned with that of MRL (Kusupati et al., 2022), as detailed in Section 3.3. Specifically, we apply a Top$k$ operator with varying values of $k$ to the backbone's output embedding and finetune the model using LoRA (Hu et al., 2022). We restrict $k$ to powers of two (i.e., $k \in \{2^i\}$), and assign a uniform weight of 1 to each $k$-dimensional sub-embedding during training. The finetuning objective is the InfoNCE loss (Oord et al., 2018), and the selection of hyperparameters is provided in Table 6.

Table 6: Implementation details on backbone finetuning in text representation.

| Backbone | Batch Size | LoRA $r$ | LoRA $\alpha$ | lr | epoch | warmup | weight decay | Top$k$ |
|---|---|---|---|---|---|---|---|---|
| e5-Mistral-7B | 256 | 8 | 16 | 2e-5 | 10 | 1000 | 0.1 | $\{1, 2, ..., 2048, 4096\}$ |
| Qwen3-Embedding-4B | 256 | 8 | 16 | 2e-5 | 10 | 1000 | 0.1 | $\{1, 2, ..., 2048, 2560\}$ |

In the training of CSRv2, we adopt the tied encoder–decoder architecture as proposed in CSR (Wen et al., 2025). For the $k$-annealing schedule, the initial sparsity level $k_{\text{init}}$ is set to 64 if the current number of activated dimensions $k$ is less than 64; otherwise set $k_{\text{init}} = 4k$. Positive and negative

---

[4]https://huggingface.co/spaces/mteb/leaderboard
[5]https://huggingface.co/datasets/sentence-transformers/
embedding-training-data

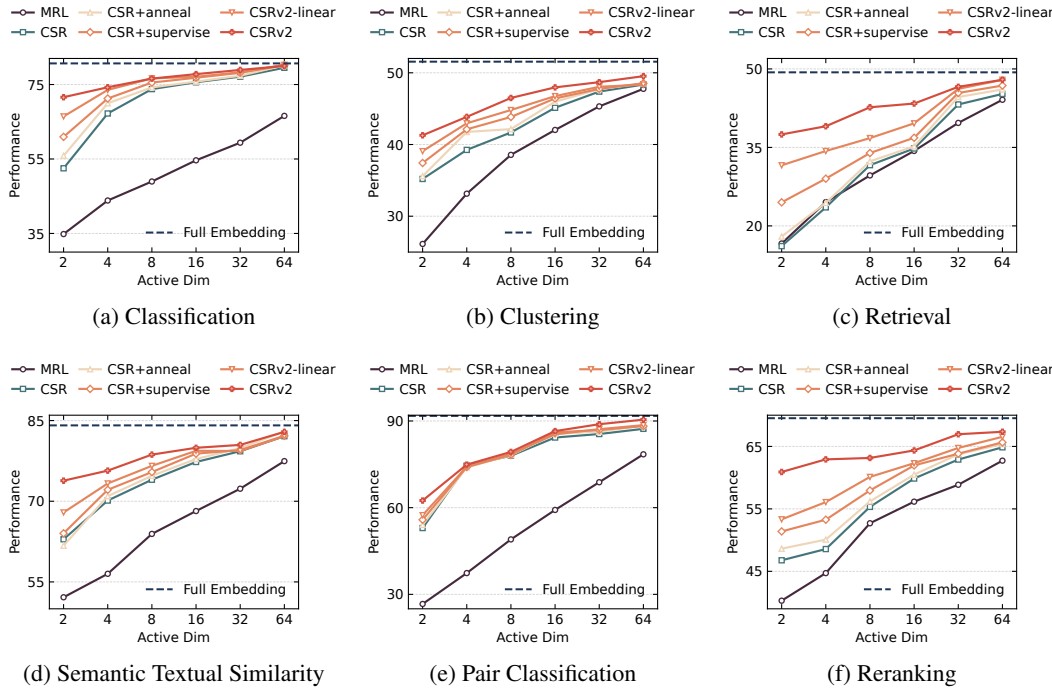

Figure 6: Task-type-specific ablation on varying components with e5-Mistral-7B as backbone.

samples for supervision are constructed in accordance with the rule detailed in Section 3.2. We employ Adam as the optimizer and selection of other hyperparameters is in Table 7.

Table 7: Implementation details on CSRv2 training in text representation.

| Backbone | d | h | lr | epoch | Batch size | $k_{aux}$ | $\beta$ | $\gamma$ | $\mathbb{K}$ | weight decay |
|---|---|---|---|---|---|---|---|---|---|---|
| e5-Mistral-7B | 4096 | 16384 | 4e-5 | 10 | 128 | 1024 | 0.1 | 1 | 2,4,...,4096 | 1e-4 |
| Qwen3-Embedding-4B | 2560 | 10240 | 2e-5 | 128 | 256 | 1024 | 0.1 | 1 | 2,4,16,64,4096 | 1e-4 |

## C.4 MORE DETAILED EXPERIMENT RESULTS ON TEXT REPRESENTATIONS

Building upon the e5-Mistral-7B backbone, we further validate the efficacy of CSRv2 through an extensive performance comparison across a broad spectrum of active dimensions (ranging from 2 to 4096—the full embedding dimensionality of the backbone) and through comprehensive ablation studies.

As shown in Table 8, we report task-type-specific results for MRL, CSR, and CSRv2 across six distinct task categories.

Moreover, we systematically evaluate the impact of different component combinations on each task type and observe that the performance gains contributed by each individual component remain consistent across diverse domains. Detailed results are provided in Figure 6.

## C.5 RETRIEVAL EVALUATION COMPARISON WITH SPLADE-BASED MODELS

Table 9 demonstrates CSRv2's performance comparison with SPLADEv3 model. We select e5-Mistral-7B (Wang et al., 2023) as backbone, whose performance on MTEB retrieval tasks is on par with SPLADEv3. Note that the Active Dim $X - Y$ for SPLADEv3 means that queries have $X$ active dimensions and documents have $Y$ active dimensions, which is a common setting in LSR series model's evaluation. Results show CSRv2 is more suitable for ultra-sparse text representation generation in extreme application cases.

Table 8: Performance in task-type-specific experiments across all dimensions.

| Active Dim | Method | Classifi. ACC ↑ | Clust. V-measure ↑ | Retrieval nDCG@10 ↑ | STS Spearman ↑ | PairClassifi. AP ↑ | Rerank. MAP ↑ | Avg. |
|---|---|---|---|---|---|---|---|---|
| 4096 | e5-Mistral-7B | 80.67 | 51.55 | 49.35 | 84.11 | 91.77 | 69.52 | 69.99 |
| 2 | MRL | 34.84 | 26.13 | 16.63 | 52.14 | 26.67 | 40.30 | 33.81 |
| | CSR | 52.50 | 35.20 | 16.14 | 62.93 | 52.95 | 46.77 | 44.17 |
| | CSRv2 | 71.59 | 41.29 | 37.48 | 73.82 | 62.46 | 60.91 | 58.34 |
| 4 | MRL | 43.84 | 33.14 | 24.55 | 56.51 | 37.36 | 44.72 | 40.83 |
| | CSR | 67.22 | 39.25 | 23.54 | 70.13 | 74.55 | 48.57 | 52.94 |
| | CSRv2 | 74.26 | 43.85 | 39.04 | 75.69 | 74.90 | 62.93 | 61.01 |
| 8 | MRL | 48.95 | 38.55 | 29.65 | 63.93 | 49.02 | 52.70 | 47.09 |
| | CSR | 73.77 | 41.68 | 31.61 | 74.00 | 77.95 | 55.32 | 58.19 |
| | CSRv2 | 76.60 | 46.49 | 42.67 | 78.67 | 79.29 | 63.15 | 63.76 |
| 16 | MRL | 54.64 | 42.03 | 34.33 | 68.18 | 59.22 | 56.16 | 51.85 |
| | CSR | 75.61 | 45.12 | 34.79 | 77.30 | 84.28 | 59.86 | 61.38 |
| | CSRv2 | 77.79 | 47.97 | 43.38 | 79.94 | 86.50 | 64.36 | 65.22 |
| 32 | MRL | 59.37 | 45.31 | 39.68 | 72.33 | 68.80 | 58.86 | 56.37 |
| | CSR | 77.11 | 47.38 | 43.21 | 79.30 | 85.48 | 62.90 | 64.59 |
| | CSRv2 | 78.92 | 48.69 | 46.58 | 80.48 | 88.88 | 66.95 | 66.70 |
| 64 | MRL | 66.58 | 47.76 | 44.11 | 77.46 | 78.46 | 62.72 | 61.47 |
| | CSR | 79.50 | 48.36 | 45.22 | 82.10 | 87.29 | 64.86 | 66.68 |
| | CSRv2 | 79.98 | 49.53 | 47.92 | 82.90 | 90.46 | 67.34 | 68.08 |
| 128 | MRL | 74.78 | 49.12 | 46.08 | 81.95 | 84.66 | 65.48 | 65.72 |
| | CSR | 79.70 | 49.32 | 46.68 | 82.39 | 87.83 | 65.48 | 67.34 |
| | CSRv2 | 80.14 | 49.83 | 48.27 | 83.12 | 90.75 | 67.44 | 68.32 |
| 256 | MRL | 76.52 | 49.21 | 46.64 | 82.07 | 85.25 | 66.04 | 66.37 |
| | CSR | 80.00 | 49.64 | 47.47 | 82.66 | 88.48 | 65.98 | 67.76 |
| | CSRv2 | 80.24 | 50.24 | 48.42 | 83.35 | 90.89 | 67.85 | 68.55 |
| 512 | MRL | 78.42 | 49.68 | 47.19 | 82.53 | 87.87 | 66.45 | 67.30 |
| | CSR | 80.12 | 49.86 | 47.92 | 82.97 | 88.93 | 66.51 | 68.04 |
| | CSRv2 | 80.31 | 50.65 | 48.64 | 83.50 | 91.14 | 68.30 | 68.77 |
| 1024 | MRL | 78.92 | 49.96 | 47.58 | 82.85 | 88.65 | 67.36 | 67.74 |
| | CSR | 80.26 | 50.36 | 48.16 | 83.28 | 89.67 | 67.28 | 68.41 |
| | CSRv2 | 80.50 | 50.88 | 48.82 | 83.65 | 91.41 | 68.64 | 68.97 |
| 2048 | MRL | 79.54 | 50.49 | 48.35 | 83.65 | 89.40 | 68.25 | 68.44 |
| | CSR | 80.38 | 50.79 | 48.62 | 83.63 | 90.42 | 68.36 | 68.81 |
| | CSRv2 | 80.51 | 51.27 | 48.93 | 83.88 | 91.63 | 68.87 | 69.16 |
| 4096 | MRL | 80.46 | 50.94 | 48.75 | 83.78 | 90.44 | 68.86 | 69.25 |
| | CSR | 80.54 | 51.13 | 49.13 | 83.94 | 90.99 | 68.96 | 69.16 |
| | CSRv2 | 80.49 | 51.34 | 49.16 | 83.94 | 91.70 | 69.18 | 69.25 |

# D EXPERIMENT DETAILS ON VISUAL REPRESENTATIONS

## D.1 EVALUATION METRICS

Following the methodology established by Kusupati et al. (2022), we adopt 1-nearest neighbor (1-NN) accuracy as the primary metric for evaluating visual representations. This metric is computed using FAISS (Jacob et al., 2018) with exact L2 distance search. In contrast to classification accuracy, which depends on the specific architecture and training procedure of a downstream classifier, 1-NN accuracy provides a direct assessment of whether semantically similar instances are embedded in close proximity within the representation space. Consequently, it serves as a more model-agnostic and training-free probe of intrinsic representation quality.

Table 9: CSRv2's Performance and Relative Retrieval Efficiency Comparison with SPLADEv3.

| Active Dim | Method | Retrieval Time | Arguana | CQAGaming | CQAUnix | CF-HN | Fiqa | Nfcorpus | Scidocs | Scifact | Avg. |
|---|---|---|---|---|---|---|---|---|---|---|---|
| 4096 | e5-Mistral-7B | 306.46× | 62.73 | 64.13 | 47.99 | 30.71 | 56.93 | 39.67 | 18.09 | 74.53 | 49.35 |
| 40-400 | SPLADEv3 | 27.25 × | 35.95 | 54.31 | 34.51 | 39.01 | 49.28 | 59.61 | 32.52 | 72.80 | 47.37 |
| 16-16 | SPLADEv3 | 3.63× | 29.09 | 48.76 | 29.14 | **32.54** | 35.00 | **52.78** | **30.77** | 57.22 | 39.41 |
| 16 | CSRv2 | 3.51 × | **54.98** | **59.78** | **39.17** | 26.92 | **52.07** | 33.18 | 15.56 | **65.39** | **43.38** |
| 8-8 | | 2.84 × | 21.92 | 39.74 | 21.27 | **31.86** | 28.88 | **47.59** | **26.14** | 45.09 | 32.81 |
| 4-4 | SPLADEv3 | 1.78 × | 14.71 | 28.59 | 9.62 | 22.37 | 19.25 | 28.71 | 17.43 | 28.18 | 21.11 |
| 2-2 | | 1.15 × | 6.05 | 13.84 | 3.97 | 14.53 | 9.76 | 16.28 | 7.59 | 18.73 | 11.34 |
| 2 | CSRv2 | 1.00 × | **44.86** | **53.81** | **35.74** | 18.22 | **45.27** | 29.16 | 11.97 | **60.83** | **37.48** |

## D.2 IMPLEMENTATION DETAILS

For fair comparison, we select the pretrained ResNet-50 weights, as noted in FF2048 in the MRL (Kusupati et al., 2022). Image preprocessing follows the identical pipeline employed in (Leclerc et al., 2023), (Kusupati et al., 2022) and (Wen et al., 2025). We utilize a tied encoder-decoder structure to build the CSRv2 framework and the implementation is based on Wen et al. (2025). All experiments are conducted on a server with 8 NVIDIA A100-SXM4-40GB. For backbone (FF2048) finetuning, the selection of hyperparameters is in Table 10.

Table 10: Implementation details on FF2048 finetuning in visual representation.

| Backbone | Batch Size | lr | epoch | warmup | Optimizer | weight decay | Top$k$ |
|---|---|---|---|---|---|---|---|
| FF2048 | 256 | 5e-6 | 10 | 1000 | Adam | 0.1 | $\{1, 2, ..., 2048\}$ |

For CSRv2 training, we adopt the same settings as CSR (Wen et al., 2025). In the $k$-annealing schedule, we initialize $k_{\text{init}} = 64$ if the target activated dimension $k$ is less than 64, otherwise we set $k_{\text{init}} = 4k$. For supervision, images belonging to the same semantic class are treated as positive pairs, while all others are considered negative samples. Adam is employed as the training optimizer and selection of other hyperparameters is in Table 11.

Table 11: Implementation details on CSRv2 training in visual representation.

| Backbone | d | h | lr | epoch | Batch size | $k_{\text{aux}}$ | $\beta$ | $\gamma$ | $\mathbb{K}$ | weight decay |
|---|---|---|---|---|---|---|---|---|---|---|
| FF2048 | 2048 | 8192 | 4e-5 | 10 | 4096 | 512 | 1/32 | 0.1 | 2,4,...,2048 | 1e-4 |

## D.3 1-NN CLASSIFICATION RESULTS

1-NN classification accuracy results on ImageNet-1k are shown in Table 12.

# E EXPERIMENT DETAILS ON GRAPHRAG EVALUATION

## E.1 EVALUATION METRICS

We follow the evaluation design proposed in Xiang et al. (2025). For retrieval, Context Relevance and Evidence Recall are adopted. For generation, Answer Accuracy, Faithfulness, Evidence Coverage and ROUGE-L are adopted. Detailed explanation on each metric is as follows:

- **Context Relevance(Relevance)** assesses how well the aggregate retrieved context satisfies query's semantic requirements. Higher values indicate greater fidelity between the retrieved material and the underlying informational intent of the user. Specifically, **Context**

Table 12: 1-NN accuracy of different methods on ImageNet-1k classification.

| Active Dim. | 2 | 4 | 8 | 16 | 32 | 64 | 128 | 256 | 512 | 1024 | 2048 |
|---|---|---|---|---|---|---|---|---|---|---|---|
| Full Rep. | | | | | | 71.19 | | | | | |
| MRL | 47.81 | 55.65 | 62.19 | 67.91 | 69.46 | 70.17 | 70.52 | 70.62 | 70.82 | 70.89 | 70.97 |
| CSR | 61.05 | 65.33 | 67.78 | 69.17 | 70.15 | 70.94 | 70.99 | 71.31 | 71.29 | 71.30 | 71.18 |
| CSRv2-linear | 65.78 | 67.29 | 68.42 | 69.71 | 70.39 | 71.01 | 71.11 | 71.24 | 71.23 | 71.19 | 71.19 |
| CSRv2 | 67.63 | 69.84 | 69.29 | 70.06 | 70.44 | 71.05 | 71.13 | 71.25 | 71.27 | 71.33 | 71.25 |

**Relevance** can be calculated as:

$$\textbf{Relevance} = \frac{1}{\mathcal{C}} \sum_{c \in \mathcal{C}} \text{R}(c, Q, \varepsilon) \tag{9}$$

where $\mathcal{C}$ is the set of retrieved contents, $Q$ is the query, $\varepsilon$ is the set of evidence, and operator $R$ determines whether each context $c$ is relevant to the query $Q$ and the evidence $\varepsilon$.

- **Evidence Recall(Recall)** quantifies the completeness of evidence retrieval by measuring the proportion of critical reference claims that are successfully covered by the system's output. It is defined as:

$$\textbf{Recall} = \frac{1}{|\mathcal{R}|} \sum_{c \in \mathcal{R}} \mathbf{1}(S(c, \mathcal{C})) \tag{10}$$

, where $\mathcal{R}$ is the set of reference claims, $S$ is the operator to decide whether claim $c$ is supported by the retrieved content $\mathcal{C}$ and $\mathbf{1}$ is the indicator function.

- **Answer Accuracy(ACC)** comprehensively assesses answer quality through a combination of semantic alignment and factual precision. To be specific,

$$\textbf{ACC} = \frac{1}{2}(\textbf{FC} + \textbf{SS})$$

where **FC** qualifies generation correctness and $\textbf{SS} = \cos(\mathbf{f}_i, \mathbf{c}_j)$ calculates semantic similarity.

- **ROUGE-L** calculates text similarity with n-gram overlap between generated and reference answers, capturing both syntactic and semantic alignment (Lin, 2004).

- **Faithfulness(FS)** explicitly targets hallucination risks by quantifying the proportion of generated claims that are grounded in the retrieved evidence, thereby serving as a direct measure of factual consistency between the system's output and its supporting context. It is measured as follows:

$$\textbf{FS} = \frac{|\{c \in A | S(c, C)|\}}{|A|}$$

where $A$ denotes the set of atomic claims in the proposed response, $C$ is the retrieved context and $S(c, C)$ denotes a boolean function indicating whether claim $c$ is supported by $C$.

- **Evidence Coverage(Cov)** quantifies the extent to which the generated response *incorporates* all critical evidentiary elements required to construct a comprehensive and factually complete answer. The formal computation is as follows:

$$\textbf{Cov} = \frac{|\{e \in E | M(e, G)\}|}{|E|}$$

where $E$ is the set of evidence, $G$ is the generated answer and $M(e, G)$ is a boolean function indicating whether evidence $e$ appears in the generation $G$.

## E.2 IMPLEMENTATION DETAILS

Our evaluation covers two domains proposed in Xiang et al. (2025): Medical and Novel. For fair comparison, we select Qwen3-Embedding-4B Zhang et al. (2025b) as the baseline embedding

Table 13: **CSRv2's Performance in GraphRAG-based Retrieval.** In GraphRAG-based retrieval evaluation, Qwen3-Embedding-4B is selected as backbone and two sparsity levels: 32 and 8 are selected for comparison. No data in benchmark is used in training for zero-shot evaluation.

| Embedding Model | Active Dim | Fact Retrieval | | Complex Reasoning | | Contextual Summarize | | Creative Generation | | Avg. |
|---|---|---|---|---|---|---|---|---|---|---|
| | | Recall↑ | Relevance↑ | Recall↑ | Relevance↑ | Recall↑ | Relevance↑ | Recall↑ | Relevance↑ | |
| *Medical* | | | | | | | | | | |
| Qwen3-4B | 2560 | 75.43 | 45.83 | 82.98 | 40.18 | 81.2 | 48.79 | 87.14 | 28.77 | 61.29 |
| MRL | | 48.30 | 15.05 | 63.52 | 16.06 | 53.64 | 19.38 | 84.13 | 12.20 | 39.04 |
| CSRv2-linear | 32 | 67.24 | 38.73 | 76.55 | 36.49 | 72.8 | 43.53 | 82.87 | 24.62 | 55.35 |
| CSRv2 | | **71.75** | **40.81** | **78.74** | **38.48** | **79.63** | **46.03** | **84.55** | **26.02** | **58.25** |
| MRL | | 47.01 | 8.42 | 57.86 | 9.38 | 46.49 | 8.56 | 82.64 | 4.22 | 33.07 |
| CSRv2-linear | 8 | 62.47 | 31.56 | 67.3 | 18.33 | **72.7** | **39.53** | 81.92 | 12.03 | 48.23 |
| CSRv2 | | **68.17** | **36.98** | **69.35** | **23.08** | 71.97 | 35.64 | **85.52** | **14.16** | **50.61** |
| *Novel* | | | | | | | | | | |
| Qwen3-4B | 2560 | 81.29 | 45.26 | 82.15 | 51.39 | 83.41 | 49.03 | 80.29 | 36.94 | 63.72 |
| MRL | | 68.47 | 27.91 | 72.80 | 33.48 | 76.42 | 33.22 | **78.02** | 28.36 | 52.34 |
| CSRv2-linear | 32 | 75.23 | 36.62 | 76.47 | 39.31 | 81.75 | 39.07 | 69.17 | **30.18** | 55.98 |
| CSRv2 | | **79.08** | **41.40** | **78.88** | **43.85** | **83.37** | **44.82** | 74.10 | 29.10 | **59.33** |
| MRL | | 63.20 | 19.39 | 69.71 | 22.58 | 72.08 | 22.44 | 80.82 | 20.52 | 46.34 |
| CSRv2-linear | 8 | 66.72 | 29.92 | 71.81 | 32.83 | 68.48 | 30.16 | 78.30 | 19.09 | 49.79 |
| CSRv2 | | **75.05** | **36.46** | **77.16** | **44.63** | **77.65** | **40.33** | **80.92** | **25.87** | **57.26** |

model and GPT-4o-mini for graph construction, answer generation and evaluation. Fast-graphrag (CircleMind-AI, 2025) is chosen as the GraphRAG framework, with minor change following Xiang et al. (2025) for Hugging Face Embedding support. All hyperparameters are set according to the settings in Xiang et al. (2025).

### E.3 Evaluation Results

Table 13 and 14 demonstrate CSRv2's zero-shot capability: In retrieval performance evaluation, at the same level of dimension, CSRv2 achieves performance improvements of over 15% and 7% in medical and novel domains respectively compared to MRL, while in generation accuracy evaluation, CSRv2-based systems achieve average improvements of over 10% and 3% in medical and novel domains.

## F Additional Qualitative Analysis

### F.1 Case Study of Feature Comparison between Different Methods

To facilitate a more intuitive comparison of the feature distributions induced by different representation learning methods and to elucidate the factors underlying CSRv2's substantial performance gains over both CSR and MRL, we extract two-dimensional embeddings from the IMDb dataset (Maas et al., 2011). Specifically, we obtain dense representations from MRL and ultra-sparse representations from CSR and CSRv2 under a sparsity budget of $k = 2$. The resulting embeddings are visualized via t-SNE in Figure 7, with positive and negative movie reviews rendered in green and purple respectively.

We observe that the MRL embedding demonstrates a clear separation between the majority of positive and negative reviews, reflecting its ability to capture dominant sentiment polarities. However, it exhibits notable limitations in handling compositional or contrastive sentiment expressions. For instance, in sentences such as "Although the plot of this movie is slow, the actors performed well and I really appreciated this movie", conflicting affective signals lead to ambiguous representations that cluster near the decision boundary. This suggests that dense, holistic representations may struggle to disentangle nuanced or mixed sentiment structures.

Table 14: **CSRv2's Performance in GraphRAG-based Generation.** In GraphRAG-based generation evaluation, Qwen3-Embedding-4B is selected as backbone and two sparsity levels: 32 and 8 are selected for comparison. No data in benchmark is used in training for zero-shot evaluation.

| Embedding Model | Active Dim | Fact Retrieval | | Complex Reasoning | | Contextual Summarize | | Creative Generation | | | Avg. |
|---|---|---|---|---|---|---|---|---|---|---|---|
| | | ACC↑ | ROUGE-L↑ | ACC↑ | ROUGE-L↑ | ACC↑ | Cov↑ | ACC↑ | FS↑ | Cov↑ | |
| *Medical* | | | | | | | | | | | |
| Qwen3-4B | 2560 | 61.33 | 29.65 | 69.63 | 21.67 | 72.39 | 46.19 | 69.23 | 32.04 | 37.7 | 48.87 |
| MRL | 32 | 45.30 | 19.88 | 55.65 | 16.69 | 55.17 | 30.65 | 64.08 | 25.11 | 34.04 | 38.51 |
| CSRv2-linear | | 52.82 | 25.03 | 61.36 | 19.02 | 64.14 | 39.73 | 66.97 | 28.45 | 35.33 | 43.65 |
| CSRv2 | | **60.69** | **29.27** | **68.60** | **20.76** | **71.18** | **45.74** | **68.44** | **31.58** | **36.68** | **48.10** |
| MRL | 8 | 35.16 | 12.64 | 47.90 | 12.99 | 41.84 | 20.04 | 57.23 | 18.89 | 29.13 | 30.65 |
| CSRv2-linear | | 49.65 | 24.48 | 57.07 | 16.49 | 59.45 | 33.82 | **69.80** | 28.17 | 34.24 | 41.46 |
| CSRv2 | | **58.09** | **27.43** | **65.21** | **19.44** | **68.83** | **41.69** | 66.47 | **29.91** | **36.07** | **45.90** |
| *Novel* | | | | | | | | | | | |
| Qwen3-4B | 2560 | 57.02 | 31.76 | 54.63 | 19.67 | 70.62 | 47.85 | 59.70 | 44.51 | 38.53 | 47.14 |
| MRL | 32 | 45.72 | 25.65 | 45.06 | 18.23 | 65.85 | 43.78 | 57.38 | 31.28 | **36.82** | 41.09 |
| CSRv2-linear | | 51.26 | 28.49 | 49.02 | 18.68 | 67.03 | 44.42 | 57.71 | 35.18 | 35.79 | 43.06 |
| CSRv2 | | **54.69** | **31.63** | **51.47** | **19.49** | **68.19** | **45.67** | **57.87** | **37.41** | 35.89 | **44.70** |
| MRL | 8 | 39.51 | 22.47 | 42.23 | 16.25 | 59.64 | 37.71 | 54.56 | 29.39 | 34.45 | 37.36 |
| CSRv2-linear | | 48.78 | 25.13 | 46.75 | 17.03 | 63.84 | 41.12 | **57.23** | 34.08 | **35.96** | 41.10 |
| CSRv2 | | **52.94** | **29.25** | **50.93** | **18.92** | **67.54** | **44.80** | 56.45 | **36.86** | 34.49 | **43.58** |

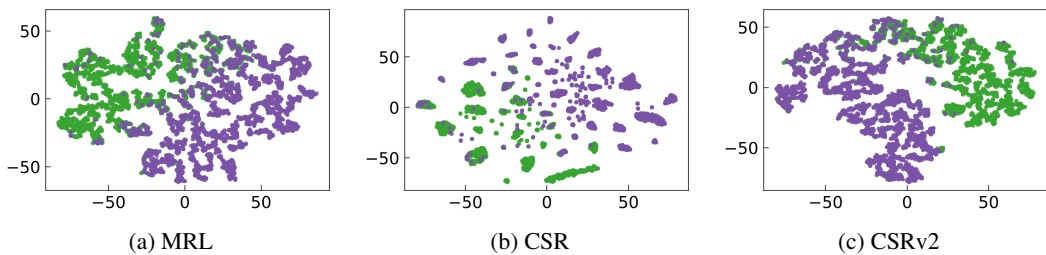

(a) MRL      (b) CSR      (c) CSRv2

Figure 7: t-SNE visualization of 2-dimensional features in IMDb generated by MRL, CSR and CSRv2 with e5-Mistral-7B as backbone. The AP scores of MRL, CSR and CSRv2 are respectively 89.34%, 92.75% and 94.62%.

In contrast, CSR adopts a compositional strategy by decomposing sentiment into fine-grained semantic primitives. This yields a highly fragmented latent space in the reduced two-dimensional projection, characterized by numerous small, localized clusters. While many of these clusters correspond to lexically precise phrases (e.g., "I really like" or "fail to"), the model faces ambiguity when encountering polysemous terms, such as "strong" which appears in both positive and negative contexts. Consequently, representations involving such terms are scattered across disparate clusters, undermining feature consistency and increasing the risk of misclassification due to unstable feature binding.

CSRv2 mitigates this issue by jointly optimizing sparse feature learning with supervised signals that promote the emergence of both high-level sentiment abstractions (e.g., overall positivity or negativity) and the fine-grained semantic patterns preserved in CSR. Crucially, we observe that individual neurons in CSRv2 consistently activate in response to emotionally salient yet lexically general terms, such as "awful" and "fantastic", which exhibit strong sentiment polarity while retaining broad contextual applicability. This hybrid inductive bias enables CSRv2 to achieve a more robust and interpretable separation of sentiment classes, effectively balancing semantic specificity with generalization capacity.

### F.2 Auto-interpretability Study on CSRv2 Neurons under Different Compression Settings

We analyze the semantic roles of individual neurons in the CSRv2 latent space under two sparsity regimes $k = 64$ (moderately sparse) and $k = 2$ (extremely sparse) using the IMDb dataset (Maas et al., 2011). For each neuron, we compute its activation values across all input sentences and identify the top-10 paragraphs that elicit the strongest responses. To interpret the semantic and affective patterns encoded by each neuron, we leverage Qwen-7B-Chat (Bai et al., 2023) to generate concise summaries of the linguistic and emotional characteristics common to these maximally activating sentences.

Our analysis reveals that under the $k = 64$ regime, while many neurons encode semantically coherent and sentiment-relevant concepts, a non-negligible subset predominantly activates in response to high-frequency yet functionally neutral lexical items, such as "I" or "today", which carry little to no emotional polarity. In contrast, under extreme sparsity ($k = 2$), neuron activations exhibit markedly increased specialization: each active dimension consistently aligns with a distinct sentiment pole, either positive or negative. This indicates that ultra sparsity constraints exert strong pressure on the model to prioritize emotionally salient, task-relevant signals, thereby yielding representations that are not only more polarized but also more interpretable in terms of their affective semantics.

## G Empirical Analysis

### G.1 Efficiency Analysis Details

Our efficiency analysis focuses on **retrieval and storage**, where computational cost meaningfully differs across methods. Even though end-to-end latency, encoder latency, and index construction cost could be relevant in a fully online setting, in most practical scenarios where embeddings are applied to downstream tasks, **pre-caching** is inevitable. That is, the corpus is encoded once, and embeddings are stored for repeated use. Typical examples include (1) **RAG systems**, where documents change infrequently and their embeddings serve millions of queries, and (2) **online services** such as recommendation, where real-time encoding of large-scale text is infeasible. Therefore, encoder and index construction costs are amortized and do not dominate real-world latency. To ensure fair comparison, we keep the encoder and indexing pipeline identical for all baselines (MRL, CSR, and CSRv2), so that any efficiency or performance variation arises strictly from the embedding representations.

With Qwen3-Embedding-4B (Zhang et al., 2025b) as the backbone, we record encoding time on a 1M corpus sampled from MTEB retrieval and reranking datasets. Table 15 shows that CSRv2 introduces only negligible overhead compared to MRL (0.012% extra time, $\sim$19.172s in total), which is insignificant relative to the hours required for large-scale corpus encoding.

Table 15: Encoding time comparison on a 1M corpus.

| Method | Encoding Time (s) |
|--------|-------------------|
| MRL    | 159854.091        |
| CSR    | 159876.478        |
| CSRv2  | 159873.263        |

In contrast, retrieval and storage costs differ dramatically. Under a fixed encoder and index type, the dominant factor in retrieval time is effective embedding dimensionality ($d$ for dense baselines vs. $k$ for CSRv2). As shown in Table 16, ultra-sparse vectors yield up to $7\times$ faster retrieval than dense MRL and up to $300\times$ speedup over the uncompressed backbone on a 1M-scale corpus. Retrieval times are averaged over 2000 rounds (batch size 512), excluding 100 warm-up iterations.

These results reinforce our main practical claim: CSRv2 offers substantial gains in the components that dominate real-world latency (retrieval throughput and embedding storage), while incurring negligible overhead on the encoder side.

Table 16: Retrieval time per query under different active dimensions (ms).

| Method | 2 | 4 | 8 | 16 | 64 | 4096 |
|---|---|---|---|---|---|---|
| MRL | 1.402 | 1.428 | 1.571 | 1.748 | 3.972 | 68.522 |
| CSRv2 | 0.227 | 0.370 | 0.633 | 0.797 | 3.217 | 45.722 |

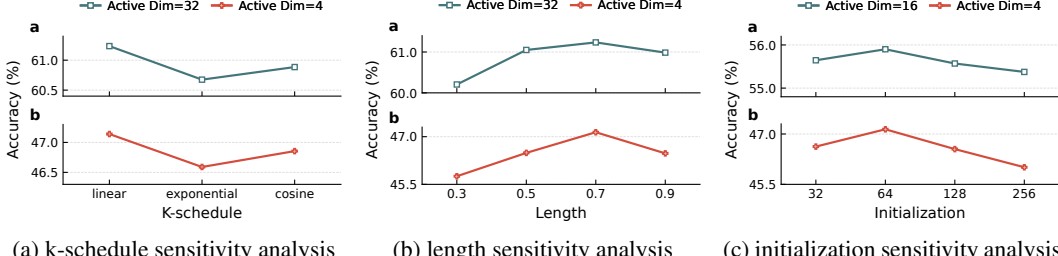

(a) k-schedule sensitivity analysis    (b) length sensitivity analysis    (c) initialization sensitivity analysis

Figure 8: **K-annealing sensitivity analysis**. **(Left)**: Sensitivity on three k-schedule strategies: linear, exponential and cosine. **(Middle)**: Sensitivity on four annealing lengths. **(Right)**: Sensitivity on four different initializations.

## G.2 K-ANNEALING SENSITIVITY ANALYSIS

We evaluate k-annealing strategy's sensitivity from three perspectives: k-schedule, length and initialization. For k-schedule, we adopt three settings: linear, exponential and cosine. For length, we take four settings: 0.3, 0.5, 0.7 (in the main paper) and 0.9. For initialization, we take four settings: 32, 64, 128 and 256. Evaluation are done in two MTEB task types (classification and retrieval) and two active dimensions are selected for each experiment for generalization. Results in Figure 8 demonstrate that different k-schedule results in relatively stable increase in performance improvement, while our selected settings: $k$ initialized to 64, annealing to target sparsity level at 70% step, and linear-annealing strategy achieves the best performance.

## G.3 ANALYSIS ON UNBALANCED WEIGHTABLE SETTINGS

MRL (Kusupati et al., 2022) has explored the impact of different weightage settings for smaller representation sizes. They find that on ImageNet (Deng et al., 2009), setting larger weight on dimensions 8 and 16 result in 3% improvement on $d = 8$, with minor performance degradation on larger dimensions. Therefore, we further conduct additional studies into the impact of imbalanced weighting.

We propose **MRL-reweight**, where we follow MRL's settings and set the following weights $\{5, 4, 3, 2, 1, \ldots, 1\}$ for dimensions 2, 4, 8, 16, 32, and so on, up to 4096. As shown in Table 17, applying larger weights at earlier stages does, to some extent, improve the performance of MRL on low-dimensional scales. However, while MRL-reweight offers some improvements, the performance does not quite reach the level of CSRv2. We hypothesize that this discrepancy arises because sparse vectors, which are more comprehensive in capturing feature combinations, are more difficult to achieve with a truncated representation (e.g., retaining only the first 8 values).

## G.4 DISCUSSION ON MULTI-SCALE LOSS TERMS

The core idea behind k-annealing curriculum is that setting larger $k_{\text{init}}$ promotes exploration and diverse neuron activations. This raises a question: can annealing in the curriculum be replaced by multiple terms that cover the range from $k_{\text{init}}$ to $k_{\text{final}}$, rather than simply setting reconstruction loss as $L(k) + \frac{1}{8}L(4k)$? We conduct quantitative discussion on two cases to look deeper into this problem:

- **Multi-TopK loss over diverse $k$s:** Since covering all $k$s (e.g., 64 loss terms) along the annealing would be too computationally prohibitive, we now consider a diverse representative subset that we use for evaluation: $k \in [2, 4, 8, 16, 32, 64]$.

Table 17: Performance comparison between standard MRL and MRL-reweight.

| Method | 2 | 4 | 8 | 16 | 32 | 64 | 128 | 256 | 512 | 1024 | 2048 | 4096 |
|---|---|---|---|---|---|---|---|---|---|---|---|---|
| | | | | | | *Classification* | | | | | | |
| MRL | 34.84 | 43.84 | 48.95 | 54.64 | 59.37 | 66.58 | 74.78 | 76.52 | 78.42 | 78.92 | 79.54 | 80.46 |
| MRL-reweight | 45.08 | 52.42 | 55.18 | 61.97 | 65.73 | 70.82 | 76.61 | 77.93 | 77.18 | 77.56 | 79.28 | 80.42 |
| CSRv2 | 37.48 | 39.04 | 42.67 | 43.38 | 46.58 | 47.92 | 48.27 | 48.42 | 48.64 | 48.82 | 48.93 | 49.16 |
| | | | | | | *Retrieval* | | | | | | |
| MRL | 16.63 | 24.55 | 29.65 | 34.33 | 39.68 | 44.11 | 46.08 | 46.64 | 47.19 | 47.58 | 48.35 | 48.75 |
| MRL-reweight | 24.48 | 29.75 | 33.25 | 37.56 | 42.12 | 46.17 | 47.20 | 46.96 | 46.82 | 47.24 | 48.17 | 48.37 |
| CSRv2 | 37.48 | 39.04 | 42.67 | 43.38 | 46.58 | 47.92 | 48.27 | 48.42 | 48.64 | 48.82 | 48.93 | 49.16 |

- **CSRv2 with start/end multi-TopK**. In this variant, we keep only the boundary losses (i.e. $k_{\text{init}} = 64$ and $k_{\text{final}} = 2$).

We compare the performance of different methods on the MTEB classification and retrieval subsets and also report their corresponding training costs. Results in Table 18 and Table 19 demonstrate that focusing on start and end Ks help a bit on addressing ultra-sparsity but there is still a large gap to our $k$-annealing. Moreover, better $k$ coverage with diverse multi-TopK delivers larger gains, but it still underperforms $k$-annealing, while introducing significant training overhead. Therefore, we believe that $k$-annealing is more preferable than these static multi-TopK loss variants. It would be interesting to look deeper into their interplay in future work.

Table 18: Performance comparison with static multi-scale loss terms on MTEB Classification and Retrieval using e5-Mistral-7B.

| Method | Active Dim | Classification | Retrieval |
|---|---|---|---|
| CSR | | 52.50 | 16.14 |
| CSRv2-linear-StartEndTopK | 2 | 57.46 | 23.65 |
| CSRv2-linear-DiverseMultiTopK | | 61.75 | 24.18 |
| CSRv2-linear-anneal | | 66.43 | 31.58 |

Table 19: Training time comparison with static multi-scale loss terms on MTEB Classification and Retrieval using e5-Mistral-7B

| Time (s) | Classification | Retrieval |
|---|---|---|
| CSRv1 | 271.32 | 638.77 |
| CSRv2-linear-StartEndTopK | 285.94 | 653.13 |
| CSRv2-DiverseMultiTopK | 501.18 | 1183.36 |
| CSRv2(anneal) | 274.15 | 642.65 |

# H FURTHER DISCUSSIONS

## H.1 EMERGENCE OF SUPERCLASS SEPARABILITY UNDER ULTRAHIGH SPARSITY

Past works (Fallah et al., 2020) have shown that sparse codes are argued to induce disentangled, semantically meaningful features. However, a key open question remains: when the sparsity is extremely high (i.e., very few active dimensions), do such representations still preserve higher-level semantic structure (such as superclasses or domains), or do they collapse into trivial, instance-specific separations?

We conduct a superclass-level analysis on two multi-intent classification datasets, Banking77 (Casanueva et al., 2020) and MTOPIntent (Li et al., 2020). For Banking77, following the semantic

structure commonly adopted in prior work, we group its 77 types of bank-related queries into 8 semantically coherent superclasses (e.g. account&identity, card management. For MTOPIntent, we adopted the original intent taxonomy in the paper and grouped these intents into 11 domains (e.g. alarm, music). These groupings allow us to evaluate whether ultrahigh sparsity induces representations that align with higher-level semantic partitions.

We evaluated MRL, CSR, and CSRv2 under the ultra-sparse regimes of $k = 2$ and $k = 4$, which correspond to ultra-sparse setting. Evaluation is done in accordance with MTEB benchmark, where a logistic regression is trained on the training set and evaluated on the test set. Results in Table 20 demonstrate a consistent and notable trend: CSRv2 produces significantly more structured sparse representations than CSR and MRL, even under extremely low $k$. Superclass clusters become more linearly separable under CSRv2, indicating that ultrahigh sparsity does not degrade semantic abstraction.

Table 20: Performance comparison for superclass classification with Qwen3-Embedding-4B as backbone.

| Method | Active Dim | Banking77 (original-class) | MTOP (original-class) | Banking77 (super-class) | MTOP (super-class) |
|---|---|---|---|---|---|
| MRL | 2 | 3.57 | 5.16 | 28.96 | 37.08 |
| CSR | 2 | 11.75 | 18.08 | 77.43 | 83.26 |
| CSRv2 | 2 | **17.03** | **23.52** | **88.44** | **93.16** |
| MRL | 4 | 6.93 | 11.51 | 31.04 | 45.24 |
| CSR | 4 | 19.02 | 24.39 | 82.91 | 86.79 |
| CSRv2 | 4 | **23.16** | **28.51** | **94.43** | **97.56** |

## H.2 Quantized Comparison at Fixed Memory Cost

To provide a more holistic view of the efficiency-accuracy trade-off landscape, we further evaluate CSRv2 of different levels of precision under fixed bit size in three MTEB task types: classification, clustering and retrieval. We take two fixed bit sizes (64 and 128), and adopt three quantization (FP32, BF16, binary) settings under each bit size.

Table 21: Performance comparison on CSRv2 and dense MRL in fixed memory cost.

| Method | Bit Size | Quantization | Active Dim | Classification | Clustering | Retrieval |
|---|---|---|---|---|---|---|
| CSRv2 | 64 | FP32 | 2 | 71.59 | 41.29 | 37.48 |
| | | BF16 | 4 | 73.05 | 42.46 | 38.19 |
| | | binary | 64 | 74.12 | 44.53 | 40.28 |
| | | PQ | 64 | 62.39 | 33.16 | 21.76 |
| MRL | 64 | binary | 64 | 64.48 | 40.04 | 27.61 |
| | | PQ | 64 | 58.37 | 37.18 | 22.04 |
| CSRv2 | 128 | FP32 | 4 | 74.26 | 43.85 | 39.04 |
| | | BF16 | 8 | 75.02 | 44.76 | 40.98 |
| | | binary | 128 | 76.30 | 45.01 | 42.26 |
| | | PQ | 128 | 70.15 | 38.97 | 30.17 |
| MRL | 128 | binary | 128 | 72.54 | 44.37 | 29.15 |
| | | PQ | 128 | 68.42 | 41.58 | 25.61 |

Results in Table 21 demonstrate that CSRv2 remains highly competitive across a wide range of quantization strategies. The findings further indicate that (1) increasing the number of active di-

mensions is often more advantageous than raising numerical precision, and (2) extremely compact binary variants of CSRv2 yield the strongest accuracy–memory trade-offs. Notably, CSRv2–binary also substantially outperforms binary-quantized dense embeddings, implying that structured sparsity provides greater representational expressiveness than uniform quantization when bit budgets are extremely constrained. Together, these observations underscore that CSRv2 constitutes a flexible and efficient embedding mechanism capable of adapting to both moderate- and ultra-low-bit compression regimes.

In addition, the consistently strong performance of binary and higher-dimensional BF16 variants suggests that richer or more varied activation patterns can effectively compensate for the semantic degradation introduced by low numerical precision. This highlights a promising direction for CSR-style representations: exploiting larger or more structured sparse activation patterns to further enhance expressiveness under increasingly aggressive quantization settings.

We also evaluate PQ on both baseline and CSRv2 embeddings with code budget 64. We use standard PQ settings with 256 codewords per subspace and 8 subvectors, and for 128-bit codes into 16 subvectors. For CSRv2, we apply PQ quantization on TopK=256's embedding. However, PQ does not outperform Binary Quantization (BQ) in this context. We attribute this performance gap to a fundamental structural mismatch. Standard PQ partitions vectors into independent subspaces with equal bit-budgets, implicitly assuming a uniform distribution of semantic information. However, MRL and CSRv2 embeddings are strictly hierarchical, concentrating "core" semantics in the early/sparse dimensions. Consequently, PQ's uniform allocation strategy disrupts this hierarchy by inefficiently assigning equal capacity to both the highly informative prefix dimensions and the fine-grained tail dimensions, resulting in suboptimal quantization. Conversely, Binary Quantization preserves the sign information of high-value dimensions directly, offering superior compatibility with hierarchical representations.

### H.3 POTENTIAL APPLICATIONS OF CSRv2 IN VECTOR QUANTIZATION

Vector quantization (VQ) methods, including Product Quantization (PQ) (Jegou et al., 2010), Optimized Product Quantization (Ge et al., 2013), and more recent anisotropic schemes such as AVQ (Guo et al., 2020), are central to real-world large-scale vector search systems where memory footprint, latency, and hardware efficiency are critical. While our main work focuses on the role of ultra-sparse representations in improving retrieval quality and compute efficiency, it is worth noting that CSRv2 is highly compatible with these widely-used quantization techniques.

CSRv2's ultra-sparse structure, activating only $k \in \{2, 4, 8\}$ dimensions out of a large latent space, naturally complements vector quantization methods such as PQ and AVQ. As only a few coordinates are non-zero, quantization can be applied exclusively to these active values (or their indices), enabling a two-stage compression pipeline of **sparsity + quantization** that substantially reduces both memory and lookup cost. Unlike dense embeddings (e.g., MRL), where quantization error spreads across all dimensions, CSRv2 concentrates signal in a handful of features, making the quantization process more signal-preserving and aligned with anisotropic quantization principles. This compatibility also facilitates integration into practical ANN systems (e.g., DiskANN (Jayaram Subramanya et al., 2019)) that already combine graph-based search with PQ, suggesting that CSRv2 can further lower system-level memory while maintaining high recall. However, as discussed in Appendix H.2, while PQ presents an interesting avenue, it necessitates adaptation to function effectively with MRL/CSRv2's hierarchical representations, which we leave for future work.

### H.4 LIMITATIONS OF CSRv2 ON THE MOST EXTREME SETTING

Although CSRv2 achieves strong performance under ultra-sparse regimes, it suffers notable degradation in the most extreme sparsity setting ($k = 1$), which reduces the representation to a hard clustering assignment. As shown in Table 22, activating only a single neuron still yields a significant improvement over baseline methods; however, CSRv2's performance drops by 27.56% relative to the dense backbone model. This decline is more than twice as severe as the degradation observed when $k = 2$ (11.65%).

Apart from those discussed in Section 5, another hypothesis for this sharp performance drop stems from the complete absence of *feature combination* when only one latent dimension is active. With

a single activation, the model loses the capacity to compose multiple semantic cues—a capability that has been shown to be critical for robust representation learning under sparsity constraints (Gao et al., 2024). Potential remedies for this limitation may involve architectural innovations that enable richer single-feature representations, such as nonlinearly compositional encoders (Li et al., 2025) or hierarchical autoencoders that preserve multi-level semantic structure even under extreme sparsity levels. Exploring such directions remains a promising avenue for future work, leaving room for future improvement.

Table 22: Performance comparison at the most extreme setting $k = 1$.

| Active Dim | Method | Classifi. AP ↑ | Clust. V-measure ↑ | Retrieval nDCG@10 ↑ | STS Spearman ↑ | PairClassifi. AP ↑ | Rerank. MAP ↑ | Avg. |
|---|---|---|---|---|---|---|---|---|
| 4096 | e5-Mistral-7B | 80.67 | 51.55 | 49.35 | 84.11 | 91.77 | 69.52 | 69.99 |
| 1 | MRL | 17.52 | 14.70 | 3.81 | 37.93 | 12.95 | 23.98 | 19.52 |
| | CSR | 28.54 | 24.14 | 6.54 | 48.93 | 37.96 | 28.16 | 28.79 |
| | CSRv2-linear | 39.61 | 28.78 | 19.82 | 51.80 | 43.85 | 37.08 | 36.63 |
| | CSRv2 | 52.43 | 31.48 | 24.73 | 54.46 | 47.09 | 42.34 | 42.43 |

## H.5 ANALYSIS ON ONE PROMISING SAE-VARIANT

Recently there have been a variant SAE called MRL-SAE (Bussmann et al., 2025) that combines MRL's core idea into SAE training. Specifically, a standard SAE whose single encoder–decoder is trained to act as many nested autoencoders at once. The encoder produces one sparse feature vector, but the decoder is forced to reconstruct the input from multiple truncations of that vector (e.g., first 256, 512, . . . , 4096 latents), and the loss is the sum of these reconstruction errors plus sparsity. This simple change to the training objective induces a hierarchy where early latents encode broad, reusable features and later latents add increasingly fine-grained detail, all within one overcomplete dictionary.

We compare MRL-SAE's performance in classification, clustering and retrieval tasks in MTEB with vanilla SAE and CSR. Results in Table 23 shows that MRL-SAE underperforms vanilla SAE and CSR for embedding tasks and also suffer from severe degradation in sparse representation generation.

Table 23: Performance comparison on MRL-SAE, vanilla SAE and CSR.

| Method | Active Dim | Classification | Clustering | Retrieval |
|---|---|---|---|---|
| vanilla SAE | | 76.74 | 46.85 | 42.09 |
| MRL-SAE | 32 | 76.49 | 46.45 | 41.57 |
| CSR | | 77.11 | 47.38 | 43.21 |
| vanilla SAE | | 72.95 | 40.27 | 30.43 |
| MRL-SAE | 8 | 72.33 | 39.19 | 29.74 |
| CSR | | 73.77 | 41.68 | 31.61 |

## I LLM USAGE STATEMENT

In accordance with the ICLR policy, we disclose the utilization of Large Language Models (LLMs) in the preparation of this manuscript. The application of these tools was strictly confined to linguistic and formatting support. Specifically, an LLM was employed to proofread the text, correct grammatical errors, and enhance the clarity and readability of the prose. The LLM played no role in any substantive scientific components of this work, including the conception of research ideas, the design of methodologies, the execution or analysis of experiments, or the generation of results and conclusions. All intellectual contributions and the essential content of this paper are exclusively attributable to the authors.

