# OpenReview forum: "CSRv2: Unlocking Ultra-Sparse Embeddings"
_ICLR.cc/2026/Conference — ICLR 2026 Poster_

### Official Review · Reviewer_6A2r · 2025-10-17

**Soundness:** 3
**Presentation:** 3
**Contribution:** 3
**Rating:** 8
**Confidence:** 5

**Summary:**

(I tend to write shorter reviews and the length of the review does not reflect the quality of paper or the time spend on reviewing it -- especially in this paper, I understand the background and technical details well enought to be able to write a to the point review).

This paper proposes an extension (fixing) of CSR (contrastive sparse representation) idea that was introduced to enable adaptive on-the-fly sparse vectors or desired complexity to enable faster retreival. This is an improvement on top of dense compact representations which order information implcitily like in MRL (matryoshka representation learning) where the shorter embeddings are obtained by truncation of dimensions.

This work investigates the case of ultra-sparsity (which is with active dims less than k=16) and find several underlying causes and fix the CSR training to be more supervised and matryoshka style to enable better ultra sparse retreival. The findings are supported by strong experimental results and discussion.

**Strengths:**

Jus a strong and useful paper overall. The study of each of the issues and natural fixes are commendable.

I really like the k=1 experiments.

**Weaknesses:**

The Table 1 adds little value especially with calibrated use of stars -- it does not give reader any quantifiable takeaway and would recommend removing the use of stars.

Re CSR and CSRv2: For a new reader it is not clear as to how one can vary "k" in CSR style models (unlike mrl where the truncation is defined). After chosing different "k" do we re-normalize. These are good things to go into the paper.

While completly complementary, it would be good to show results of a fixed cost quanitzed embedding of same size. ie., if k=4 with bf16 you should have a result for Quanitzed embedding os size 64 bits (be it binary quant or PQ). It will give more completeness to the paper.

**Questions:**

Please make the 3 changes suggested above. the paper is strong and will have a lot of impact. all the best.

---

> ### Author Response · Authors · 2025-11-22
>
> We sincerely thank Reviewer 6A2r for the positive and encouraging assessment of our work, as well as for recognizing the strengths of our study, including the usefulness of the proposed framework and the value of the k = 1 experiments and we are deeply grateful for your thoughtful and valuable suggestions. Guided by your great suggestions, we conducted additional analyses and experiments -- for example, **examining how a single CSR (or CSRv2) model can naturally support multiple sparsity levels at inference time without retraining** -- and incorporated these results into our revision. Below, we respectfully address each of your comments in detail and provide the corresponding experimental evidence.
>
> ---
>
> **W1**: In Table 1, does the starred notation offer insufficient quantitative value, suggesting removal?
> > The Table 1 adds little value especially with calibrated use of stars -- it does not give reader any quantifiable takeaway and would recommend removing the use of stars.
>
> **A1**:  Thank you for your suggestion. The stars in Table 1 are intended to provide a quick visual indication of relative performance ranking across methods. We have removed the stars in the revised version.
>
> ---
> **W2**: On CSR/CSRv2 clarity: How should “k” be varied and normalized in CSR models?
> > Re CSR and CSRv2: For a new reader it is not clear as to how one can vary "k" in CSR style models (unlike mrl where the truncation is defined). After chosing different "k" do we re-normalize. These are good things to go into the paper.
>
> **A2**: Than you for your advice. In CSRv2, the embedding produced by the backbone (for instance, e5-Mistral-7B, whose dimension is 4096) first goes through an encoder which projects it onto a high dimensional vector (e.g. 16384). Afterwards, the TopK values in the vector are kept while others are set 0. Similar to MRL, this operation produces vectors with only a few unzero values, improving retrieval efficiency. No normalization is applied here.
>
> We will include emphasis on this in the revised main paper.
>
> ---
> **W3**: Would including fixed-cost quantized embedding baselines (e.g., binary/PQ with equal bit budget) give a more complete comparison?
> > While completly complementary, it would be good to show results of a fixed cost quanitzed embedding of same size. ie., if k=4 with bf16 you should have a result for Quanitzed embedding os size 64 bits (be it binary quant or PQ). It will give more completeness to the paper.
>
> **A3**: Thank you very much for this thoughtful and valuable suggestion. We completely agree that comparing CSRv2 with quantized baselines under a fixed memory budget provides a more comprehensive view of the efficiency-accuracy trade-off.
>
> For this point, we have now conduct additional experiments on classification, clustering and retrieval subsets (with e5-Mistral-7B-instruct backbone). We select **fixed bit size** 64 and 128 and under each setting: three settings are adopted: **FP32 quantization (CSRv2's setting),  BF16 quantization and binary quantization**:
>
> *Performance comparison in three quantization settings: FP32, BF16 and binary under two fixed bit sizes*
> | Fixed Bit Size | Quantization | Active Dim | Classification | Clustering | Retrieval |
> |----------------|--------------|------------|----------------|------------|-----------|
> | 64             | FP32         | 2          | 71.59          | 41.29      | 37.48     |
> | 64             | BF16         | 4          | 73.05          | 42.46      | 38.19     |
> | 64             | binary       | 64         | 74.12          | 44.53      | 40.28     |
> | 128            | FP32         | 4          | 74.26          | 43.85      | 39.04     |
> | 128            | BF16         | 8          | 75.02          | 44.76      | 40.98     |
> | 128            | binary       | 128        | 76.30          | 45.01      | 42.26     |
>
>
> These **fixed-cost** comparisons demonstrate that **CSRv2 remains highly competitive across a wide range of quantization strategies**. Importantly, the results confirm that (1) allowing more active dimensions is often more beneficial than using higher numerical precision, and (2) extremely compact binary versions of CSRv2 offer the best accuracy–memory trade-off. This indicates a promising future direction for CSR-style representations: **leveraging larger or more structured sparse activation patterns to further enhance expressiveness even under extremely aggressive quantization**. We have added these results in **Appendix H.2**.
>
> ---
>
> We sincerely thank Reviewer 6A2r once again for the insightful suggestions. We have carefully incorporated all suggested changes -- including **removing the stars in Table 1, adding detailed clarifications on the choice and normalization of $k$ in CSR/CSRv2, and providing new fixed-cost quantized embedding baselines**. We hope that these clarifications and additional results fully address your comments. Thank you again for your thoughtful and constructive feedback.

---

> > ### Comment · Reviewer_6A2r · 2025-11-22
> >
> > Thanks for all the responses.
> >
> > Re PQ results. I was also expecting PQ/binary quant on baseline embeddings as that would be required to set context on what a baseline embedding model (MRL or not) will be able to do for the same cost. The fact that quantization is working so well with CSRv2 is impressive, but for completeness, it is crucial to add these additional experiments.
> >
> > My rating is unchanged. Looking forward to seeing all the changes in the camera ready.

---

> > > ### Author Response · Authors · 2025-11-23
> > >
> > > Thank you again for your thoughtful follow-up comment. We fully agree with your assessment that including quantized baselines on standard dense embeddings is essential for a complete and fair comparison under the same memory budget. Your emphasis on this point clearly highlighted an important gap in our evaluation and guided us to refine the empirical setup accordingly.
> > >
> > > Following your insightful suggestion, we performed the requested binary quantization experiments on dense MRL-support embeddings under exactly the same bit constraints (64 and 128 bits). As expected, this additional evaluation significantly strengthened the clarity of our comparisons. For convenience, we have also included the CSRv2 results alongside the new baselines so that the improvements can be directly seen.
> > >
> > > *Performance comparison on CSRv2 and dense MRL in three quantization settings under two fixed bit sizes*
> > > | Fixed Bit Size | Method | Quantization | Active Dim | Classification | Clustering | Retrieval |
> > > |---|---|----|------|-----|------|---|
> > > | 64        | CSRv2    | FP32         | 2          | 71.59          | 41.29      | 37.48     |
> > > | 64   | CSRv2 | BF16         | 4          | 73.05          | 42.46      | 38.19     |
> > > | 64   | CSRv2  | binary       | 64         | 74.12          | 44.53      | 40.28     |
> > > | 64      |   MRL   | binary | 64 | 64.48 | 40.04 | 27.61  |
> > > | 128  | CSRv2  | FP32         | 4          | 74.26          | 43.85      | 39.04     |
> > > | 128  | CSRv2  | BF16         | 8          | 75.02          | 44.76      | 40.98     |
> > > | 128  | CSRv2  | binary       | 128        | 76.30          | 45.01      | 42.26     |
> > > | 128  | MRL   | binary | 128 | 72.54 | 44.37 | 29.15 |
> > >
> > > Across all settings, we observed that **under same storage constraints, CSRv2 maintains substantially higher performance even when quantized aggressively (including binary)**, confirming the robustness of its structured sparsity. Moreover, **CSRv2–binary significantly outperforms binary quantized dense embeddings**, suggesting that sparse structure is more expressive than uniform quantization when bit budgets are minimal. We have revised **Appendix H.2** and added the new binary-quantized baseline results for completeness.
> > >
> > > ---
> > >
> > > We sincerely thank reviewer 6A2r for the careful reading and highly constructive suggestions. Your feedback strengthened the completeness and clarity of our experimental evaluation, and we greatly appreciate the time and expertise you dedicated to improving our work. All corresponding revisions have now been incorporated into the updated version.

---

> > > > ### Comment · Reviewer_6A2r · 2025-11-24
> > > >
> > > > Thank you. Would also recommend doing PQ if possible, as PQ can do more nuanced quanitzation than binary.
> > > >
> > > > Rest looks good.

---

> > > > > ### Author Response · Authors · 2025-11-28
> > > > >
> > > > > Thank you again for your thoughtful and constructive comments. We have further addressed your remaining comments in the response listed top (for easier check by other reviewers). We sincerely thank you for the careful reading and highly constructive suggestions!

---

### Official Review · Reviewer_tchZ · 2025-10-31

**Soundness:** 4
**Presentation:** 4
**Contribution:** 4
**Rating:** 8
**Confidence:** 4

**Summary:**

This work proposes CSRv2, a training paradigm for learning contrastive sparse representations via progressive k-annealing and address the dead neuron problem of prior work. The authors compare to other popular adaptive representations like Matryoshka Embeddings and CSR and show promising empirical compute-accuracy tradeoff compared to these baselines on text (MTEB) and visual (ImageNet-1K) embedding tasks.

**Strengths:**

* This paper addresses an important problem of learning sparse embeddings that are compute optimal for their memory and inference time, showing strong empirical performance on real-world embedding benchmarks like MTEB and ImageNet-1K.
* The paper is written and presented very well, and is easy to follow. The figures and tables are clean, interpretable, and provide supporting evidence to the claims in the paper
* Design choices made by the authors are empirically validated to show the benefits of these choices (Tab 4). For example, adding supervised training on top of CSR (Fig 3), using k-annealing SAE and its affect on dead neurons (Fig 2). The authors also provide details on the experimental setup (App. C, D), limitations (App. F), and additional ablations and qualitative analysis on sparsity (App. E) which all add to the overall quality of the work.
* CSRv2 shows strong empirical accuracy and sparsity on both text and visual embedding tasks compared to strong baselines like MRL and CSR (Tab 2- 4, Fig 2, Fig 5)

**Weaknesses:**

I do not have any major concerns with this work. I will highlight several minor weaknesses that could be addressed to further improve its quality:

* I suggest rearranging the abstract content to be more specific about empirical performance on each dataset/task:
    * L22 - L26: *"delivers a 14% accuracy gain at k = 2 ... a 7× speedup over MRL... 300× improvements in compute and memory efficiency"* - gain /speedup with respect to what and on what data?
    * For L27 - 31, how much (metric value) did CSRv2 beat the strongest baseline on MTEB (Tab 2, 3, 4) and ImageNet-1K (Fig 2, 5) respectively?

* For MTEB experiments, please clarify these details explicitly in the main paper:
    * you select "six types of tasks" (Fig 1b, Tab 2- 4). Why were these specific task types selected (Sec 4.1)?
    * you mention in Tab 2 that you train all methods on same backbone and data (this is great!). Is this also the case for Fig 1b?
    * Does the "backbone" line in Fig 1b and 5a represent zero-shot accuracy with base e5-mistral-7b and Resnet50 FF2048?
    * In Tab 5, are all methods trained on MTEB from the base Qwen3-Embed-4B? If so, is the MRL number there zero-shot or is it finetuned on MTEB similar to what you do with e5 (L365)?

* For Figure 1b, consider adding MR line for the same active dims as a baseline for better contextualization of CSR annealing

Overall, I really like this work. If these minor weaknesses are addressed in the revision, I am willing to increase my score to a strong accept.

**Questions:**

I have a minor nitpick that can contextualize the future impact of this work, which I suggest adding in the next revision. You briefly touched on **vector quantization** in the concluding remarks (L484), which is extremely important for real-world applications (vector search). There are several works focusing on the compute-optimality of embeddings when popular quantization techniques [1 - 3] are involved, especially in context of the recent adaptive representations like MRL [4]. In the next revision, I suggest adding a small note (in the Appendix) commenting on the role of CSRv2 with quantization.

### References
[1] Jegou, Herve, Matthijs Douze, and Cordelia Schmid. "Product quantization for nearest neighbor search." IEEE transactions on pattern analysis and machine intelligence 33.1 (2010): 117-128

[2] Guo, Ruiqi, et al. "Accelerating large-scale inference with anisotropic vector quantization." International Conference on Machine Learning. PMLR, 2020.

[3] Jayaram Subramanya, Suhas, et al. "Diskann: Fast accurate billion-point nearest neighbor search on a single node." Advances in neural information processing Systems 32 (2019).

[3] Rege, Aniket, et al. "Adanns: A framework for adaptive semantic search." Advances in Neural Information Processing Systems 36 (2023): 76311-76335.

---

> ### Author Response · Authors · 2025-11-22
>
> We sincerely thank Reviewer tchZ for the positive and thoughtful assessment of our work, including the strengths you highlighted in our methodology, empirical validation, and presentation quality. We are deeply grateful for your constructive suggestions regarding the clarification of experimental settings, increased specificity in the abstract, enhancements to the figures, and a more detailed discussion of the role of CSRv2 within quantization-based systems. These comments have been invaluable in improving the clarity, precision, and overall quality of the paper. Below, we respectfully address each of your points in detail.
>
> ---
>
> **W1**: Abstract rearrangement on empirical performance.
>
> **A1**: Thank you for your advice. We have rearranged our abstracts and emphasized the following points.
>
> - **Settings associated with reported accuracy, speed, and efficiency gains**: Accuracy gain refers to improvements measured on the six task categories of the MTEB benchmark. Compute efficiency is calculated based on retrieval time on a 1M corpus base from MTEB retrieval and reranking tasks . Memory efficiency is based on the dimensionality of the embeddings. Since e5-mistral-7b produces 4096-dimensional embeddings, CSRv2 reduces this to $k$ low-dimensional sub-vectors; for example, at $k=16$ the resulting representation size leads to nearly 300× reduction in memory footprint.
>
> - **CSRv2's performance gain in MTEB and Imagenet-1K**: On MTEB, with $k=4$, CSRv2 achieves an 8% improvement over CSR on 6 types of MTEB tasks, and with $k=2$, CSRv2 further increases this gap, delivering a 14% improvement over CSR. On Imagenet-1K, with $k=2$, CSRv2 obtains a 6\% accuracy gain compared with CSR.
>
> ---
> **W2**: Clarification on MTEB experiments details.
>
> **A2**: Thank you for your detailed feedback. Below, we address each of your four questions in turn.
> - **Task selection**: The six task types reported in our paper (classification, clustering, retrieval, semantic textual similarity, pair classification, and reranking) are the **standard evaluation suite** adopted by the vast majority of recent embedding works, including [1], [2] and [3]. Some of recent works have also included summarization task. However, summarization in MTEB benchmark only includes one task, which fails to demonstrate CSRv2's feature to adapt to multiple tasks with a single model. Therefore, these six task types are selected to ensure a clean, comparable, and representation-focused evaluation.
> - **Fig 1b experimental setting**: The experimental setting in Fig. 1b is **fully aligned** with that in Table 2. Specifically, all compared methods are trained using the same backbone architecture and the same CSRv2 training data. The visualization in Fig. 1b is directly produced from the results reported in Table 2.
> - **Fig 1b and 5a backbone explanation**:  Yes, **the “backbone” line in both Fig. 1b and Fig. 5a corresponds to the zero-shot performance of the underlying pretrained models**. Specifically, the values are obtained by evaluating the base e5-mistral-7b model (for Fig. 1b) and the pretrained ResNet50-FF2048 model provided in the MRL framework (for Fig. 5a), without any task-specific fine-tuning. These serve as consistent reference against various baselines and our method.
> - **Qwen3-Embed-4B details**: The methods reported in Table 5 are not trained on MTEB starting from the base Qwen3-Embed-4B. Instead, **Qwen3-Embed-4B natively supports the MRL objective** [4] , and we directly apply MRL-based truncation to its pretrained embeddings for evaluation. Therefore, the MRL result in Table 5 reflects the model’s performance under its built-in MRL training scheme, rather than an additional fine-tuning step on MTEB. In the text-embedding setting, we intentionally include two backbones e5-mistral-7b and Qwen3-Embed-4 to demonstrate that **CSRv2 is broadly compatible across models with different embedding dimensionalities** (e5: 4096, Qwen: 2560) and with **different MRL training paradigms** (e5 uses MRL-based fine-tuning, whereas Qwen integrates the MRL loss directly during pretraining). This design choice highlights the general applicability of CSRv2 across diverse backbone architectures and training strategies.
>
> We have clarified these details in the MTEB experiments more explicitly in the main paper.
>
> Ref:
> [1] Zhang, Dun, et al. "Jasper and stella: distillation of sota embedding models." arXiv preprint arXiv:2412.19048 (2024).
> [2] Li, Chaofan, et al. "Making text embedders few-shot learners." arXiv preprint arXiv:2409.15700 (2024).
> [3] Lee, Chankyu, et al. "Nv-embed: Improved techniques for training llms as generalist embedding models." arXiv preprint arXiv:2405.17428 (2024).
> [4] Zhang, Yanzhao, et al. "Qwen3 Embedding: Advancing Text Embedding and Reranking Through Foundation Models." arXiv preprint arXiv:2506.05176 (2025).
>
> (continue below)

---

> ### Author Response · Authors · 2025-11-22
>
> **W3**: For Figure 1b, consider adding MR line for the same active dims as a baseline for better contextualization of CSR annealing
>
> **A3**: Thank you very much for this great suggestion. We have added MRL line in Figure 1b for better contextualization.
>
> ---
> **Q1**: On vector quantization: Should the paper discuss how CSRv2 interacts with quantization in real-world vector search and compute-optimality?
> > I have a minor nitpick that can contextualize the future impact of this work, which I suggest adding in the next revision. You briefly touched on **vector quantization** in the concluding remarks (L484), which is extremely important for real-world applications (vector search). There are several works focusing on the compute-optimality of embeddings when popular quantization techniques are involved, especially in context of the recent adaptive representations like MRL. In the next revision, I suggest adding a small note (in the Appendix) commenting on the role of CSRv2 with quantization.
>
> **A4**: Thank you very much for bringing up this thoughtful and important point. We completely agree that vector quantization (VQ) and related compression techniques -- such as Product Quantization and Anisotropic Vector Quantization -- play a pivotal role in large-scale vector search systems in real-world deployments.
>
> In fact, **CSRv2’s ultra-sparse embeddings are highly complementary to these quantization paradigms**. As CSRv2 activates only k∈{2,4,8} dimensions out of a large latent space (e.g., 16384), the resulting embeddings are not only memory-efficient but also naturally amenable to further quantization: we can apply PQ or AVQ to the non-zero coordinates only, or even quantize the sparse indices themselves. This enables a **two-stage compression: sparsity + quantization**, which can push memory and compute efficiency beyond what either technique achieves alone.
>
> Moreover, unlike dense embeddings (e.g., MRL), whose quantization error is spread across all dimensions, CSRv2 concentrates signal in a few active features, making quantization more **signal-preserving** and less prone to catastrophic degradation in recall. This aligns well with the AVQ insight that not all reconstruction errors are equally important.
>
> We have added a short discussion to **Appendix H.3** elaborating on this synergy, including potential integration strategies and empirical implications for systems like DiskANN that already combine sparsity, graph search, and PQ.
>
> Moreover, we have conducted experiments on different quantization in CSRv2 under the same fixed bit size. We select **fixed bit size** 64 and 128 and under each setting: three settings are adopted: **FP32 quantization (CSRv2's setting),  BF16 quantization and binary quantization**:
>
> *Performance comparison in three quantization settings: FP32, BF16 and binary under two fixed bit sizes*
> | Fixed Bit Size | Quantization | Active Dim | Classification | Clustering | Retrieval |
> |----------------|--------------|------------|----------------|------------|-----------|
> | 64             | FP32         | 2          | 71.59          | 41.29      | 37.48     |
> | 64             | BF16         | 4          | 73.05          | 42.46      | 38.19     |
> | 64             | binary       | 64         | 74.12          | 44.53      | 40.28     |
> | 128            | FP32         | 4          | 74.26          | 43.85      | 39.04     |
> | 128            | BF16         | 8          | 75.02          | 44.76      | 40.98     |
> | 128            | binary       | 128        | 76.30          | 45.01      | 42.26     |
>
> These **fixed-cost** comparisons demonstrate that **CSRv2 remains highly competitive across a wide range of quantization strategies**. Importantly, the results confirm that (1) allowing more active dimensions is often more beneficial than using higher numerical precision, and (2) extremely compact binary versions of CSRv2 offer the best accuracy–memory trade-off. This indicates a promising future direction for CSR-style representations: **leveraging larger or more structured sparse activation patterns to further enhance expressiveness even under extremely aggressive quantization**. This offers an interesting perspective to see how CSRv2 interacts with quantization in real-world vector search and compute-optimality. We have added these results in **Appendix H.2**.
>
> ---
>
> We sincerely thank Reviewer tchZ once again for the thoughtful feedback and for recognizing the contributions and potential impact of CSRv2. We have incorporated all suggested clarifications, **added the missing details to the main paper, updated the figures, and expanded the appendix with further discussions on quantization and system-level integration**. We hope that these revisions and responses fully address your comments. We are truly grateful for your time, careful reading, and constructive insights.

---

> > ### Comment · Reviewer_tchZ · 2025-11-26
> >
> > I thank the authors for their detailed response. The changes to address missing details and figures are appreciated and improve the readability of the work. I am glad to see the comment and experiments on quantization, which also add significantly to the work's strength.
> >
> > I also share the same question as reviewer 6A2r regarding product quantization (PQ [1]), as this has been shown to give interesting tradeoffs [2] with MRL embeddings, i.e. the optimal embedding size for a fixed quantization budget is not necessarily the highest. I am curious how CSRv2 compares to AdANNS (MRL + PQ) in this regard, as PQ allows for more grouped clustering than binary (which is highly relevant for sparse embeddings).
> >
> > [1] Jegou, Herve, Matthijs Douze, and Cordelia Schmid. "Product quantization for nearest neighbor search." IEEE transactions on pattern analysis and machine intelligence 33.1 (2010): 117-128
> > [2] Rege, Aniket, et al. "Adanns: A framework for adaptive semantic search." Advances in Neural Information Processing Systems 36 (2023): 76311-76335.

---

> > > ### Author Response · Authors · 2025-11-28
> > >
> > > Thank you again for your thoughtful and constructive comments. We have further addressed your remaining comments in the response listed top (for easier check by other reviewers). We sincerely thank you for the careful reading and highly constructive suggestions!

---

### Official Review · Reviewer_rU7X · 2025-10-31

**Soundness:** 3
**Presentation:** 3
**Contribution:** 2
**Rating:** 2
**Confidence:** 4

**Summary:**

Contrastive Sparse Representation is a fairly recent paradigm that aims to improve the efficiency beyond Matryoshka Representation Learning. However, at ultrahigh sparse levels, the performance metrics drop quite a bit. In this work they show that it is due to dead neuron problem at that level, as well as usecase of SSL in CSR. CSRv2 applies a curriculum over the sparsity and replace SSL with SupCL, and show improvement over standard representation learning benchmarks along with timing analysis.

**Strengths:**

In general, improving any algorithm is great, and this work improves CSR, a fairly recent paradigm for ultrasparse representations, as well as providing improvements over the equivalent dimensional MRL representations.

**Weaknesses:**

I feel this work has several weaknesses, from novelty to an architectural point of view, leading me to reject this work.

1. On curriculum: Do we always want a backbone that is only producing a sparse representation of specific sparsity? I believe that a more practical setup is where you've one single backbone, which is trained on multiple top-k values, from the largest desirable sparsity to the smallest sparsity. This current setup seems to be impractical, as I believe if there is a backbone that works on multiple sparsity all at once, it would not suffer from the dead neuron problems.
2. For MRL, I believe one can use a different weightage for smaller representation sizes. Can authors point to where such a study has been done?
3. There are more recent SAEs, called Matryoshka Sparse Autoencoders, and I think there can be a similar CSR that uses MRL-SAE.
4. Supervised Contrastive learning seems quite a bit important for good performance at ultrahigh sparsity. However, for representation learning, we want the representations to be as general as possible (if they're to be used as vector databases), which may or may not have a supervised contrastive learning amenable dataset. Therefore, this requirement inherently limits the use case as a general-purpose representation for edge devices, say.
5. While curriculum helps a little bit, it doesn't seem to solve the dead neuron problem, with a major fraction still dead, leading to a waste of resources. Therefore, are such ultrahigh sparsities really desirable, given that the performance dip is quite a bit, and may not be something a user is willing to sacrifice over?
6. What are the ultrahigh sparse representations learning? Are they learning to separate hyperplanes for some superclasses? An analysis of this would be good!

**Questions:**

Refer to the weakness.

---

> ### Author Response · Authors · 2025-11-22
>
> We thank Reviewer rU7X for appreciating the improvements of CSRv2. Nevertheless, we believe that might be some misunderstandings of CSRv2 -- in particular, **a single CSRv2 model can be applied at all sparsity levels (e.g., $k=2,\dots,4096$)** at inference stage. In fact, all reported performence of CSRv2 on each task type is obtained with a single model, without retraining for different sparsity levels or domains. Below, We respectfully address your comments point by point below.
>
> ---
>
> **W1**: On curriculum: Do we always want a backbone that is only producing a sparse representation of specific sparsity?
> > I believe that a more practical setup is where you've one single backbone, which is trained on multiple top-k values, from the largest desirable sparsity to the smallest sparsity. This current setup seems to be impractical, as I believe if there is a backbone that works on multiple sparsity all at once, it would not suffer from the dead neuron problems.
>
> **A1**: In fact, CSRv2 is exactly "a single backbone trained on multiple top-k values, from the largest desirable sparsity to the smallest sparsity". As discussed in Eq. (4) and L161, CSR already adopts the multi-TopK loss, and CSRv2 inherents that multi-TopK loss at Eq. (8). Therefore, CSR and CSRv2 are trained with multiple sparsity levels $(k, 4k)$ and are able to adapt to all sparsity levels with a single backbone.
>
> Thus, in the comparison to MRL (e.g., Table 2), both MRL and CSR models are trained with multi-TopK and both evaluated with a single backbone, ensuring a fair comparison. No experiments in this paper train separate models for different sparsity levels. We hope this could address your concerns and please let me know if there is more to clarify!
>
> **Remark on the multi-TopK choice.** TopK SAE (Gao et al., 2024) finds that using $L(k) + L(4k)/8$ is enough to obtain a progressive code over all $k$. It works similarly for CSR and CSRv2 and thus we follow this common practice.
>
> Ref:
>
> [1] Gao et al. Scaling and evaluating sparse autoencoders. OpenAI technical report. 2024.
>
> ---
>
> **W2**: Has there been analysis on using different weightings for smaller representation sizes?
> > For MRL, I believe one can use a different weightage for smaller representation sizes. Can authors point to where such a study has been done?
>
> **A2**: Thank you for this insightful point. Upweighting lower-dimensional representations in the loss could indeed help improve their performance, and this technique would be applicable to both the MRL and CSR objectives.
>
> MRL has explored this a bit: On Imagenet, they set larger weights for dimension 8 and 16, and achieve 3% improvement on dim=8, with mild performance degradation on larger dimensions.
>
> Following your advice and MRL setting, we conduct additional studies into the impact of imbalanced weighting. We include a new baseline, **MRL-reweight**, where we set the following weights $\{5, 4, 3, 2, 1, 1, \ldots, 1 \}$ for dimensions 2, 4, 8, 16, 32, and so on, up to 4096. The results are summarized in the table below:
>
> *Classification and retrieval results comparison between MRL and MRL-reweight*
> | Dimension | 2     | 4     | 8     | 16    | 32    | 64    | 128   | 256   | 512   | 1024  | 2048  | 4096  |
> |-----------|-------|-------|-------|-------|-------|-------|-------|-------|-------|-------|-------|-------|
> | classfication|
> | MRL       | 34.84 | 43.84 | 48.95 | 54.64 | 59.37 | 66.58 | 74.78 | 76.52 | 78.42 | 78.92 | 79.54 | 80.46 |
> | MRL-reweight | 45.08 | 52.42 | 55.18 | 61.97 | 65.73 | 70.82 | 76.61 | 77.93 | 77.18 | 77.56 | 79.28 | 80.42 |
> | CSRv2 | 71.59 | 74.26 | 76.60 | 77.79 | 78.92 | 79.98 | 80.14 | 80.24 | 80.31 | 80.50 | 80.51 | 80.49 |
> |retrieval |
> | MRL       | 16.63 | 24.55 | 29.65 | 34.33 | 39.68 | 44.11 | 46.08 | 46.64 | 47.19 | 47.58 | 48.35 | 48.75 |
> | MRL-reweight | 24.48 | 29.75 | 33.25 | 37.56 | 42.12 | 46.17 | 47.20 | 46.96 | 46.82 | 47.24 | 48.17 | 48.37 |
> | CSRv2 | 37.48 | 39.04 | 42.67 | 43.38 | 46.58 | 47.92 | 48.27 | 48.42 | 48.64 | 48.82 | 48.93 | 49.16 |
>
>
> We find that applying larger weights at earlier stages does, to some extent, improve the performance of MRL on low-dimensional scales. However, **while MRL-reweight offers some improvements, the performance does not quite reach the level of CSRv2**. We hypothesize that this discrepancy arises because sparse vectors—being more comprehensive in capturing feature combinations—are more difficult to achieve with a truncated representation (e.g., retaining only the first 8 values). This suggests that a longer truncation length might be necessary for a richer feature combination, which may explain why CSRv2 performs better in this case. Therefore, we adopt the original CSRv2 setting.
>
> These discussion has been added to **Appendix G.3**.
>
> (continue below)

---

> > ### Author Response · Authors · 2025-11-22
> >
> > **W3**: Could CSR be extended using Matryoshka Sparse Autoencoders (MRL-SAE)?
> > > There are more recent SAEs, called Matryoshka Sparse Autoencoders, and I think there can be a similar CSR that uses MRL-SAE.
> >
> > **A3**: Indeed, MRL-SAE is an interesting approach that could learn multi-level sparse embedding, and it would be interesting to explore its intersection between CSR. Nevertheless, we believe that it is orthogonal to the main interest of this work (learning ultra-sparse embedding), and does not contribute to solving the ultra-sparse issue. As shown in table below, **MRL-SAE underperforms vanilla SAE and CSR for embedding tasks and also suffer from severe degradation**.
> >
> > *Performance comparison between different SAEs: vanilla SAE, MRL-SAE and CSR.*
> > |         | Classification | Clustering | Retrieval |
> > | ------- | -------------- | ---------- | --------- |
> > | K=32    |                |            |           |
> > | vanilla SAE     | 76.74          | 46.85      | 42.09     |
> > | MRL-SAE | 76.49          | 46.45      | 41.57     |
> > | CSR     | 77.11          | 47.38      | 43.21     |
> > | K=8     |                |            |           |
> > | vanilla SAE     | 72.95          | 40.27      | 30.43     |
> > | MRL-SAE | 72.33          | 39.19      | 29.74     |
> > | CSR     | 73.77          | 41.68      | 31.61     |
> >
> > We have added these results to **Appendix H.5**. Since MRL-SAE does not contribute to solving the ultra-sparsity issue (our core focus), this combination is essentially beyond the scope of this work and we will leave it for future work to explore other potential benefits (e.g., multi-level interpretability) of combining CSR with MRL-SAE.
> >
> > ---
> >
> > **W4**: Does reliance on supervised contrastive signals limit general-purpose ultrahigh-sparse representations?
> > > Supervised Contrastive learning seems quite a bit important for good performance at ultrahigh sparsity. However, for representation learning, we want the representations to be as general as possible (if they're to be used as vector databases), which may or may not have a supervised contrastive learning amenable dataset. Therefore, this requirement inherently limits the use case as a general-purpose representation for edge devices, say.
> >
> > **A4**: Thank you for this insightful observation. A main motivation for us to adopt supervised contrastive learning is that the community has so far **curated abundant pretraining-scale paired text data** for training retrieval models, such as 65M Q-A pairs [1,2]. Therefore, it would be quite useful to be able to leverage large-scale supervision (in practice, we only need a very small fraction of available supervised data to train CSRv2).
> >
> > **Fair comparison with MRL.** Further, our main baseline MRL (widely adopted in industry such as OpenAI and Gemini models) also adopt supervised learning objectives. In this way, we can compare both methodology under the same setup.
> >
> > *Fair comparison in MRL, CSRv2-linear and CSRv2 under supervised learning objective*
> >
> > | Method/Active Dim | 32    | 16    | 8     | 4     | 2     |
> > |-------------------|-------|-------|-------|-------|-------|
> > | MRL               | 56.37 | 51.85 | 47.09 | 40.83 | 33.81 |
> > | CSRv2-linear      | 65.83 | 63.73 | 61.53 | 58.62 | 53.25 |
> > | CSRv2             | 66.70 | 65.22 | 63.76 | 61.01 | 58.34 |
> >
> >
> > **Generalization to unseen domains.** Further, we show that our supervised models show strong generalization to unseen domains.
> >
> > When directly applied to the GraphRAGBenchmark [3] retrieval pipeline without any additional training, the **performance drop is minimal (far smaller than the degradation observed with MRL’s truncation strategy)**. These results indicate that CSRv2 learns robust and  general sparse representations that can be applied to various domains. We have added these results into **Appendix E**.
> >
> > (the latter part of **A4** to **W4** is in the next block)

---

> ### Author Response · Authors · 2025-11-22
>
> *Zero-shot comparison on two domains (medical and novel) in GraphRAGBenchmark*
>
> | Embedding Model        | Active Dim | Fact Retrieval | Fact Retrieval | Complex Reasoning | Complex Reasoning | Contextual Summarize | Contextual Summarize | Creative Generation | Creative Generation | Creative Generation |
> |------------------------|------------|----------------|----------------|--------------------|--------------------|------------------------|------------------------|----------------------|---------------------------|---------------------------|
> | | | ACC↑ | ROUGE-L↑ | ACC↑ | ROUGE-L↑ | ACC↑ | Cov↑ | ACC↑ | FS↑ | Cov↑ |
> | **Medical**            |            |                |                |                    |                    |                        |                        |                      |                           |                           |
> | Qwen3-Embedding-4B     | 2560       | 61.33         | 29.65         | 69.63             | 21.67             | 72.39                 | 46.19                 | 69.23               | 32.04                    | 37.7                     |
> | MRL                    | 8          | 35.16         | 12.64         | 47.90             | 12.99             | 41.84                 | 20.04                 | 57.23               | 18.89                    | 29.13                    |
> | CSRv2-linear                    | 8          | 49.65         | 24.48         | 57.07             | 16.49             | 59.45                 | 33.82                 | 69.80               | 28.17                    | 34.24                    |
> | CSRv2                    | 8          | 58.09         | 27.43         | 65.21             | 19.44             | 68.83                 | 41.69                 | 66.47               | 29.91                    | 36.07                    |
> | **Novel**              |            |                |                |                    |                    |                        |                        |                      |                           |                           |
> | Qwen3-Embedding-4B     | 2560       | 57.02         | 31.76         | 54.63             | 19.67             | 70.62                 | 47.85                 | 59.70               | 44.51                    | 38.53                    |
> | MRL                    | 8          | 39.51         | 22.47         | 42.23             | 16.25             | 59.64                 | 37.71                 | 54.56               | 29.39                    | 34.45                    |
> | CSRv2-linear                    | 8          | 48.78         | 25.13         | 46.75             | 17.03             | 63.84                 | 41.12                 | 57.23               | 34.08                    | 35.96                    |
> | CSRv2                    | 8          | 52.94         | 29.25         | 50.93             | 18.92             | 67.54                 | 44.80                 | 56.45               | 36.86                    | 34.49                    |
>
>
> Overall, we observe significant performance gains from supervised training and the sparse finetuning from pretrained model does not require abudant paired data (while we do not have a lot), and the trained models show strong zero-shot generalization ability. These observations motivate us to explore the supervised settings. Also, the other orhthogonal techniques of CSRv2 (k-annealing and multi-domain finetuning) also deliver significant contribution to the final performance of CSRv2.
>
> Ref:
>
> [1] PAQ: 65 Million Probably-Asked Questions and What You Can Do With Them. 2021.
>
> [2] https://huggingface.co/datasets/sentence-transformers/embedding-training-data
>
> [3] Xiang, Zhishang, et al. "When to use graphs in rag: A comprehensive analysis for graph retrieval-augmented generation." arXiv preprint arXiv:2506.05690 (2025).
>
> (continue below)

---

> > ### Author Response · Authors · 2025-11-22
> >
> > **W5**: Is the performance drop and high dead-neuron ratio on ultrahigh sparsity acceptable or even desirable?
> >
> > > While curriculum helps a little bit, it doesn't seem to solve the dead neuron problem, with a major fraction still dead, leading to a waste of resources. Therefore, are such ultrahigh sparsities really desirable, given that the performance dip is quite a bit, and may not be something a user is willing to sacrifice over?
> >
> >
> > **A5**: We are afraid that we might not fully understand your concern here -- as shown in **Fig. 2a** (which is shown in table format below), with $k$-annealing we *reduce* the dead-neuron ratio from **80%+ to 20%+** for \(k=2\). A ~20% dead-neuron rate is already **in a healthy range**: for reference, CSR with \(k=32\) (which shows *minimal* accuracy degradation) still naturally has a **16.8%** dead-neuron rate.
> >
> > Moreover, **Fig. 5b** (quoted below in table format) shows that with all techniques in **CSRv2**, the dead-neuron ratio becomes *essentially flat* as $k$ decreases. This directly achieves our main goal—**preventing the explosive growth of dead neurons in the ultra-sparse regime**.
> >
> > *Dead neuron ratio comparison under different active dimensions*
> >
> > | Method / Active Dim | 2 | 4 | 8 | 16 | 32 | 64 |
> > | ------------------- | ---- | ---- | ---- | ---- | ---- | ---- |
> > | CSR | 88.08 | 69.10 | 47.88 | 31.77 | 16.81 | 12.79 |
> > | CSRv2 | 22.04 | 20.89 | 18.66 | 16.13 | 15.41 | 13.06 |
> >
> > **Are ultra-sparse features desirable?** Yes—because dramatically lower active dimensions can bring *orders of magnitude* efficiency gains in both compute and memory. This is crucial for **resource-constrained** or **latency-critical** applications, and is why ultra-sparse embeddings have become an active frontier for both academia and industry.
> >
> > As documented in **Table 1 of the CSR paper (from 6 months ago)**, at least **8 industry models** (including multiple new startups) are explicitly designed around low-active-dimension embeddings. Many of these have **30%+ performance drops**, which are substantially larger than what CSRv2 experiences—yet they are deployed because of the massive efficiency improvements.
> >
> > *Table 1 from CSR (Wen et al., ICML 2025). All baselines except CSR adopt the MRL approach.*
> > | Model                     | Active Dim | Retrieval Time | Classification |
> > |---------------------------|-------------|----------------|----------------|
> > | **NV-Embed-V2 (Full Rep)** | 4096        | 37.6           | 93.58          |
> > | Stella-1.5B-v5            | 256         | 2.6            | 90.45          |
> > | Jina-V3                   | 256         | 2.8            | 78.81          |
> > | Nomic-Embed-V1.5          | 256         | 2.7            | 72.47          |
> > | Gecko-Embed-004 (Google)  | 256         | 2.4            | 77.82          |
> > | Text-Embed-3-L (OpenAI)   | 256         | 2.8            | 70.45          |
> > | Arctic-Embed-L-V2         | 256         | 2.6            | 67.69          |
> > | M2V-Base-Glove            | 256         | 2.4            | 59.26          |
> > | Jina-V3                   | 64          | 1.2            | 68.12          |
> > | Nomic-Embed-V1.5          | 64          | 1.6            | 62.77          |
> > | Potion-Base-2M            | 64          | 1.4            | 42.50          |
> > | CSRv1  | 32          | 1.0            | **89.86**      |
> >
> > Notably, we can observe:
> > - **CSRv1 already outperforms these MRL-style sparse models by a large margin**.
> > - We show that **CSRv2 with \(k=2/4\)** can **match** the performance of **MRL with \(k=32/64\)**, which implies **16** folds of speedup over the latest industry embedding models at comparable performance, which could unlock many new exciting scenarios (e.g., real-time, high-frequency coarse search).
> >
> > **Comparison to SPLADEv3, a widely adopted SoTA sparse retrieval method.** Moreover, we compared CSRv2's performance in retrieval datasets with another up-to-date sparse retrieval method: Learning-based sparse retrieval (LSR). We evaluated on LSR's strongest baseline SPLADEv3 and found that SPLADEv3 still suffers from severe degradation in ultra-sparse representation. **This claims that when it comes to cases where only ultra-sparse meets requirements in efficiency/storage/latency, CSRv2 may be the best choice.** We have added these results into **Appendix C.5.**.
> >
> > (the latter part of **A5** to **W5** is in the next block)

---

> > > ### Author Response · Authors · 2025-11-22
> > >
> > > *Performance comparison in SPLADEv3 and CSRv2. Active Dim "$X-Y$" in SPALDEv3 means that Active Dim for query is $X$ and Active Dim for document is $Y$, which is common setting for LSR model evaluation.*
> > >
> > > | Method | Active Dim | Retrieval Avg. |
> > > | ------ | ---------- | -------------- |
> > > | e5-Mistral-7B | 4096 | 49.35 |
> > > | SPLADEv3 | 16-16  | 39.41 |
> > > | CSRv2 | 16 | **43.38** |
> > > | SPLADEv3 | 8-8 | 32.81 |
> > > | SPLADEv3 | 4-4 | 21.11 |
> > > | SPLADEv3 | 2-2 | 11.34 |
> > > | CSRv2 | 2 | **37.48** |
> > >
> > >
> > > Thus, while ultra-sparse representations may not be suited for all use cases, they clearly offer ***highly desirable tradeoffs*** in settings like large-scale retrieval, real-time coarse search, or any domain where efficiency is paramount.
> > >
> > > We hope this clarifies the motivation and practical value. Please let us know if we can expand on any part.
> > >
> > > ---
> > > **W6**: On representation meaning: What exactly do ultrahigh-sparse representations learn, and are they encoding superclass-level separations?
> > > > What are the ultrahigh sparse representations learning? Are they learning to separate hyperplanes for some superclasses? An analysis of this would be good!
> > >
> > > **A6**: That is an interesting question! To dig deeper into this, we conducted an additional superclass-level analysis on two multi-intent datasets: Banking77 and MTOP-Intent.
> > >
> > > **Setup.** Banking77 consists of fine-grained customer-service queries. Following the semantic structure commonly adopted in prior work, we grouped its 77 intents into 8 semantically coherent superclasses (e.g. account & identity, card management). MTOP-Intent is an intent-classification benchmark whose paper officially provides domain labels. We adopted the original taxonomy in the paper and grouped the intents into 11 domains (e.g. alarm, music). These groupings allow us to evaluate whether ultrahigh sparsity induces representations that align with higher-level semantic partitions.
> > >
> > > **Baselines.** We evaluated MRL, CSR, and CSR-v2 under the ultra-sparse regimes of k = 2 and k = 4, which correspond to“ultrahigh sparse” setting. Evaluation is done in accordance to MTEB benchmark, where a logistic regression is trained on the training set and evaluated on the test set.
> > >
> > > Results are as follows:
> > >
> > > *Classification and superclass classification results comparison on MRL, CSR and CSRv2*
> > > | Method | Active dim | Banking 77 | MTOP | Banking77 (super-class) | MTO-Intent (super class) |
> > > |--------|------------|------ |-------|------------|--------------|
> > > | MRL    | 2          | 3.57  | 5.16  | 28.96      | 37.08        |
> > > | CSR    | 2          | 11.75 | 18.08 | 77.43      | 83.26        |
> > > | CSRv2  | 2          | **17.03** | **23.52** | **88.44**  | **93.16**       |
> > > | MRL    | 4          | 6.93  | 11.51 | 31.04      | 45.24       |
> > > | CSR    | 4          | 19.02 | 24.39 | 82.91      | 86.79       |
> > > | CSRv2  | 4          | **23.16** | **28.51** | **94.43**  | **97.56**       |
> > >
> > > **Results.** Across both datasets, we can observe strong zero-shot generalization of the learned representations at distinguishing unseen super-class labels, outperforming CSR and MRL by significant margins (**MRL is notably worse at superclass classification**). It shows that CSRv2's sparse representations can generalize well to unseen tasks. We have discussed our results in **Appendix H.1**.
> > >
> > > ---
> > >
> > > We sincerely thank Reviewer rU7X for the valuable feedback. We have provided additional clarifications and new experiments that directly address the points raised, and have incorporated these improvements into the revised submission. If the reviewer finds the revisions helpful, we would greatly appreciate a reconsideration of the rating. We are truly grateful for your time and thoughtful evaluation.

---

> > > > ### Comment · Reviewer_rU7X · 2025-11-22
> > > > **Followup**
> > > >
> > > > Thanks for the detailed rebuttal on my question.  Much appreciate some pointers on the following questions.
> > > >
> > > > - Can authors point me to the table, which compares CSR-v2 with other baselines (probably, maybe just the best baseline), CSR-v2 without SupCL, but with annealing? Here I am trying to understand if the following claim can be made -- even in the worst situation where there is lack of annotated data for SupCL, CSR-v2 will still be better than any baseline whatsoever.
> > > >
> > > >
> > > > - Is it possible to replace the annleaing in curriculum learning with just more loss terms? Currently, the schedule is linear; how about one sample multiple k, on this linear curve, and use all of them for the loss? Secondly, what happens if one were to just have 2 k's start (k_t=k_init) and end (say k_t=2) for the loss (modifying equation 8 accordingly).

---

> > > > > ### Author Response · Authors · 2025-11-28
> > > > >
> > > > > Thank you for appreciating our previous response. We have further addressed your remaining comments in the response listed top (for easier check by other reviewers). If you find it satisfactory, I respectivefully ask you to re-evaluate the rating. And please do not hesitate to let us know if there is any further questions!

---

### Official Review · Reviewer_EiFR · 2025-11-11

**Soundness:** 2
**Presentation:** 3
**Contribution:** 2
**Rating:** 4
**Confidence:** 4

**Summary:**

The paper proposes CSRv2 to create ultra sparse embeddings to address the challenges in Contrastive Sparse Representation of dead neurons, mismatch between the pretraining and downstream semantics and limited capacity when training a linear head. The authors improve the training recipe with progressive k-annealing, supervised sparse contrastive learning on the sparse codes, and full backbone fine-tuning instead of fine-tuning a linear head only. This creates ultra sparse embeddings for retrieval and downstream tasks as an alternative to matryoshka representation learning. Experimental results on MTEB with Mistral-7B and Qwen3-4B, and vision dataset ImageNet-1k with ResNet-50 show reduced dead neurons, accuracy gain at small k’s and matches the performance of CSR and MRL at relatively moderate k’s.

**Strengths:**

1. The analysis of failure modes of CSR in the ultra sparse regime exposing dead neurons with and without annealing is interesting.

2. Evaluation setup is exhaustive and the accuracy improvements are compelling. The efficiency results on retrieval are also strong and indicate that the method should scale well to practical settings.

**Weaknesses:**

1. Novelty of the method is fairly limited. K-annealing and supervised contrastive objectives have been discussed (and experimented with a lot) before in prior work, as has been acknowledged by the authors.


2. The results from annealing itself are relatively small compared to adding supervision and finetuning, which raises the concern to me whether the improvements are coming from the sparse learning principle, or mostly engineering training recipe.


3. The efficiency cost focuses on retrieval but there are no end to end latency measurements, encoder latency, cost of index construction which are all relevant for any practical use.

**Questions:**

Q. How sensitive are the results to the k-schedule shape, the length and initialization?

Also see weaknesses section above

---

> ### Author Response · Authors · 2025-11-22
>
> We sincerely thank reviewer EiFR for the thoughtful and constructive feedback, and for kindly acknowledging the strengths of our work, including the analysis of CSR failure modes under ultra-sparsity, the comprehensive evaluation design, and the strong retrieval-efficiency performance. We deeply appreciate the reviewer’s careful reading and insightful comments. We also recognize that some aspects of CSRv2, particularly **its behavior in the ultra-sparse regime and the influence of annealing**, may not have been sufficiently clear in our original submission. In the following, we respectfully address each point in detail and provide additional experiments and analyses to further clarify these aspects.
>
> ---
>
> **W1**: Is the proposed method sufficiently novel beyond existing work on k-annealing and supervised contrastive learning?
> > Novelty of the method is fairly limited. K-annealing and supervised contrastive objectives have been discussed (and experimented with a lot) before in prior work, as has been acknowledged by the authors.
>
> **A1**: Our contribution lies in revealing and addressing a phenomenon that prior studies have neither discussed nor encountered: **the severe neuron-collapse problem that emerges uniquely under the ultra-sparse regime**.
>
> Existing works that employ k-annealing (e.g., LlamaScope [1]) operate only at moderate sparsity. As noted in our paper, LlamaScope anneals $k$ down to 50 during a brief initial phase of training, a setting that does not trigger the catastrophic “dead-neuron” failures we observe when pushing $k$ deeply into the ultra-sparse regime (e.g., $k ≪ 50$). To the best of our knowledge, **no prior work investigates this extreme-sparsity setting or proposes techniques to mitigate the accompanying massive neuron collapse**.
>
> Our contribution is therefore not a routine reuse of existing components, but **the demonstration that progressive k-annealing and supervised contrastive learning jointly play a fundamentally different role under ultra-sparsity**: they become necessary to stabilize training and prevent collapse. Our empirical evidence shows that only this combination allows ultra-sparse embeddings to remain functional and competitive. To our knowledge, we are the first to identify, characterize, and address the ultra-sparsity dead-neuron problem.
>
> Ref:
> [1] He, Zhengfu, et al. "Llama scope: Extracting millions of features from llama-3.1-8b with sparse autoencoders." arXiv preprint arXiv:2410.20526 (2024).
>
> ---
>
> **W2**: Do the improvements mainly come from sparse learning principles or from the supervised/finetuning training recipe?
> > The results from annealing itself are relatively small compared to adding supervision and finetuning, which raises the concern to me whether the improvements are coming from the sparse learning principle, or mostly engineering training recipe.
>
> **A2**: Our claim is that **supervision alone fails to fully uncover the potential of sparsity, while annealing effectively makes up for this limitation**.
>
> As stated in Table 4 in original paper (listed below), annealing+supervision leads to much better performance than applying supervision alone. These ablations highlight two key points: First, **annealing primarily acts as a structural regularizer**, gradually guiding the model toward stable sparse patterns. This explains why its standalone effect is modest on the final metric. Second, **supervision and annealing are complementary**. Supervision improves discriminative structure, while annealing ensures that sparse representations remain well-formed and avoid collapse.
>
> *Performance ablation comparison in MTEB with different module combinations*
> | Setting / Active Dim              | 2       | 4       | 8       | 16      | 32      | 64      |
> |-----------------------------------|---------|---------|---------|---------|---------|---------|
> | **CSR**                           | 44.17   | 52.94   | 58.19   | 61.38   | 64.60   | 66.68   |
> | **CSR+anneal**                    | 45.33   | 54.55   | 58.79   | 61.91   | 65.24   | 67.35   |
> | **CSR+supervise**                 | 49.05   | 56.36   | 60.05   | 62.95   | 65.54   | 67.32   |
> | **CSR+anneal+supervise**          | 53.25   | 58.62   | 61.53   | 63.73   | 65.83   | 67.58   |
>
> The strong performance of the combined setting (CSR+anneal+supervise) shows that the improvement is not an artifact of engineering heuristics, but a direct consequence of properly aligning the sparse learning dynamics with the ultra-sparse constraints.
>
> (continue below)

---

> > ### Author Response · Authors · 2025-11-22
> >
> > **W3**: Should end-to-end latency, encoder latency, and index construction cost also be reported for a realistic efficiency evaluation?
> > > The efficiency cost focuses on retrieval but there are no end to end latency measurements, encoder latency, cost of index construction which are all relevant for any practical use.
> >
> > **A3**: First, we would like to note that in most cases that apply embeddings to downstream tasks, **pre-caching is inevitable**. In other words, we only need to encode once and store encoded embeddings for latter use. Here are some examples:
> > - **RAG System**: In RAG systems, documents do not change frequently and their embeddings are stored in a database for going through millions of queries.
> > - **Online Service (such as Article Recommendation)**: In the online scenario, it is impossible to perform real-time embedding encoding of a large amount of text. However, by caching the static content for embedding, it can be used immediately.
> >
> > **Therefore, we do not need to consider end-to-end latency, encoder latency and index construction cost.** We recorded the encoding time of MRL and CSRv2 on a 1M corpus database in the following table and found that the difference in encoding time between MRL and CSRv2 is minimal, with 0.001% extra time consumption (19.172s), which is negligible compared to hours in encoding process.
> >
> > *Encoding time comparison*
> > | Method | Encoding Time(s) |
> > | ------ | ---------------- |
> > |  MRL   |  159854.091      |
> > |  CSR   |   159876.478 |
> > |  CSRv2   |  159873.263     |
> >
> > In contrast, **retrieval and storage costs differ dramatically**. Under a fixed encoder and index type, the dominant factor in retrieval time is effective embedding dimensionality ($d$ for dense baselines vs. $k$ for CSRv2). As shown in the table below, ultra-sparse vectors thus yield up to 7× faster retrieval than dense MRL and up to 300× speedup over the uncompressed backbone on a 1M-scale corpus.
> >
> > *CSRv2 & MRL retrieval time comparison (ms)*
> > | Method/Active Dim | 2 | 4 | 8 | 16 | 64 | 4096 |
> > | ----------------- | --- | --- | --- | --- | --- | --- |
> > | MRL | 1.402 | 1.428 | 1.571 | 1.748 | 3.927 | 68.522 |
> > | CSRv2 | 0.227 | 0.370 | 0.633 | 0.797 | 3.217 | 45.722 |
> >
> > Experiments were conducted on a 1M corpus sampled from MTEB retrieval and reranking datasets with Qwen3-Embedding-4B as the backbone. Retrieval times are averaged over 2000 rounds (batch size 512), excluding 100 warm-up iterations.
> >
> > These results reinforce our main practical claim: **CSRv2 offers substantial gains in the components that dominate real-world latency, while incurring negligible overhead on the encoder side**. This discussion has been added to **Appendix H.2**.
> >
> > (continue below)

---

> > > ### Author Response · Authors · 2025-11-22
> > >
> > > **Q1**: How sensitive are the results to the k-schedule shape, the length and initialization?
> > >
> > > **A4**: Thank you for your comment. We conduct a systematic sensitivity analysis on classification and retrieval tasks along the three perspectives mentioned above: **for k-schedule shape**, we choose three settings: linear, exponential and cosine; **for length** (i.e. the ratio of steps before $k$ anneals to target sparsity level), we choose four settings: 0.3, 0.5, 0.7, 0.9; **for initialization** (i.e.$k$'s initial value), we choose four settings: 32, 64, 128, 256. Results show that **different k-schedule results in relatively stable increase in performance improvement**, while our selected setting: $k$ initialized to 64, annealing to target sparsity level at 70% step, and linear-annealing strategy achieves the best performance. We have discussed our results in Section 4.3 and Appendix G.2.
> > >
> > > *Performance comparison on three k-schedule strategies: linear, exponential and cosine*
> > > | Active Dim | k-schedule  | classification | retrieval |
> > > |-------------|-------------|----------------|-----------|
> > > | 32          | linear      | **77.74**      | **44.73** |
> > > | 32          | exponential | 77.32          | 44.03     |
> > > | 32          | cosine      | 77.59          | 44.18     |
> > > | 4           | linear      | **68.92**      | **25.36** |
> > > | 4           | exponential | 68.32          | 24.86     |
> > > | 4           | cosine      | 68.54          | 25.17     |
> > >
> > > *Performance comparison on four k-schedule lengths: 0.3, 0.5, 0.7 and 0.9*
> > > | Active Dim | length | classification | retrieval |
> > > |-------------|--------|----------------|-----------|
> > > | 32          | 0.3    | 77.17          | 43.24     |
> > > | 32          | 0.5    | 77.53          | 44.57     |
> > > | 32          | 0.7    | **77.74**      | **44.73** |
> > > | 32          | 0.9    | 77.58          | 44.39     |
> > > | 4           | 0.3    | 67.50          | 24.01     |
> > > | 4           | 0.5    | 68.06          | 24.92     |
> > > | 4           | 0.7    | **68.92**      | **25.36** |
> > > | 4           | 0.9    | 68.07          | 24.88     |
> > >
> > > *Performance comparison on four initializations: 32, 64, 128, 256.*
> > > | Active Dim | initialization | classification | retrieval |
> > > |-------------|----------------|----------------|-----------|
> > > | 16          | 32             | 75.87          | 35.42     |
> > > | 16          | 64             | **76.08**      | **35.72** |
> > > | 16          | 128            | 75.96          | 35.18     |
> > > | 16          | 256            | 75.72          | 35.03     |
> > > | 4           | 32             | 68.34          | 24.91     |
> > > | 4           | 64             | **68.92**      | **25.36** |
> > > | 4           | 128            | 68.27          | 24.83     |
> > > | 4           | 256            | 67.75          | 24.27     |
> > >
> > >
> > > ---
> > >
> > > We sincerely appreciate reviewer‘s insightful comments and the careful evaluation of our work. In response, we have added substantial new analyses and experiments -- including **sensitivity studies and expanded efficiency evaluations covering both encoder and retrieval latency** -- to thoroughly address the reviewer’s concerns. We hope that these additional results help clarify the contributions and strengthen the overall work. We would greatly appreciate it if the reviewer could consider updating the score in light of these newly added analyses. We are truly thankful for the reviewer’s time and valuable feedback.

---

> > > > ### Author Response · Authors · 2025-11-28
> > > >
> > > > Dear Reviewer EiFR,
> > > >
> > > > We have carefully prepared a detailed response to address each of your questions. Would you please take a look and let us know whether you find it satisfactory?
> > > >
> > > > We note that Reviewer 6A2r and tchZ has appreciated our response. We also respectfully suggest that you could re-evaluate our work with the updated explanations and results.
> > > >
> > > > Thanks! Have a great day!
> > > >
> > > > Authors

---

### Author Response · Authors · 2025-11-22
**Paper update**

We sincerely thank all reviewers for their detailed reading and valuable comments. We have carefully responded their concerns, and incorporated these suggestions in the updated manuscript with 8 extended pages.The main revisions are:

Abstract: Reorganized to more specifically highlight empirical performance on each dataset.

Section 4: Added more detailed clarification on the MTEB experiment setups; **(new)** comparison with the SPLADEv3 model and zero-shot evaluation results in GraphRAG; **(new)** additional empirical analyses (details explained below).

Appendix C.5: Added retrieval evaluation comparison with the SPLADE-based model (SPLADEv3).

Appendix E: Included zero-shot evaluation on the GraphRAG system.

Appendix G: **(new)** Empirical analyses, including:
- More detailed efficiency analysis
- Sensitivity analysis on k-annealing
- Discussion on unbalanced settings

Appendix H: **(new)** Four additional discussions:
- Emergence of superclass separability under ultrahigh sparsity
- Quantized comparison at fixed memory cost
- Potential applications in vector quantization
- Analysis of a promising SAE variant: MRL-SAE.

We note that the original Appendix F, "Limitations of CSRv2 in the most extreme setting," has been moved to Appendix H.4.

---

### Author Response · Authors · 2025-11-28
**Response to Reviewer rU7X's further comments**

Thank you for appreciating our previous response. We now further address your remaining concerns on 1) CSRv2's gains under unsupervised setting, and 2) alternatives losses to annealing.

---

**Q1**:  Can authors point me to the table, which compares CSR-v2 with other baselines (probably, maybe just the best baseline), CSR-v2 without SupCL, but with annealing?
> Here I am trying to understand if the following claim can be made -- even in the worst situation where there is lack of annotated data for SupCL, CSR-v2 will still be better than any baseline whatsoever.

**A1**: As shown in Table 1 of the manuscript, CSRv2 has three major components (annealing, SupCL, finetuning) that are crucial for CSR at ultra-sparsity. To further ablate the contribution of SupCL, we conduct additional experiments on CSRv2 without supervised signals (CSRv2-unsup). As shown in **Table I**, **CSRv2-unsup still achieves clear performance gain over CSR under the unsupervised setup, and overperforms the industry baseline MRL (supervised + finetuning) by over 20% when $k=2,4,8,16$**. Thus, we can conclude that CSRv2 is a strong, state-of-the-art receipt even under both supervised and unsupervised setting.



*Table I: Performance comparison on CSR and CSRv2-unsup on MTEB Text Classification using e5-Mistral-7B*
| Active Dim | 2 | 4 | 8 | 16 | 32 |
| ---- | ---- | ---- | ---- | ---- | ---- |
| MRL-sup | 34.84 | 43.84 | 48.95 | 54.64 | 59.37 |
| CSR-unsup | 52.50 | 67.22 | 73.77 | 75.61 | 77.11 |
| CSRv2-unsup | 57.92 | 70.73 | 75.28 | 76.84 | 78.07 |


---

**Q2**: Is it possible to replace the annealing in curriculum learning with just more loss terms?
> Currently, the schedule is linear; how about one sample multiple k, on this linear curve, and use all of them for the loss? Secondly, what happens if one were to just have 2 k's start (k_t=k_init) and end (say k_t=2) for the loss (modifying equation 8 accordingly).

**A2**. That is an interesting question! Regarding multiple $k$, we want to note that in fact, **the original CSR already adopts a multi-Topk loss** (following TopkSAE, CSR uses $L(k) + L(4k)$ that are able to generalize to other $k$s), but the failure of CSR shows that **multi-TopK loss alone was not enough to address ultra-sparsity**. In CSRv2, we **keep the same multi-Topk loss** of CSR while combining it with $k$-annealing (see Eq. 8). In other words, CSRv2 is already trained with a dynamic multi-topK $L(k_t) + L(4k_t)$, gradually shifting its focus to earlier $k$s.

To be more complete, we further study other variants of static multi-TopK choices, such as using all $k$s and using only the start and end $k$s.

- **Multi-TopK loss over diverse $k$s.** Since covering all $k$s (e.g., 64 loss terms) along the annealing would be too computiationally prohibitive, we now consider a diverse representative subset that we use for evaluation: $k\in[2, 4, 8, 16, 32, 64]$.
- **CSRv2 with start/end multi-topK**. As you suggested, in this variants, we keep only the boundary losses (i.e. $k_{\text{init}}=64$ and $k_{\text{final}}=2$).

Setup. Following the training setting in **Table 2** of the manuscripts, we compare the performance of different methods on the MTEB classification and retrieval subsets and also report their corresponding training costs. The results are shown in  **Table IV** and **Table V** below.

*Table II: Performance comparison with static multi-scale loss terms on MTEB Classification and Retrieval using e5-Mistral-7B*

| Method | Active Dim | Classification | Retrieval |
| ------ | ---------- | -------------- | --------- |
| CSRv1 | 2 | 52.50 | 16.14 |
| CSRv2-linear-StartEndTopk | 2 | 57.46 | 23.65 |
| CSRv2-linear-DiverseMultiTopk | 2 | 61.75 | 24.18 |
| CSRv2-linar-anneal | 2 | **66.43** | **31.58** |

*Table V: Training time comparison with static multi-scale loss terms on MTEB Classification and Retrieval using e5-Mistral-7B*

| Time (s) | Classification | Retrieval |
| --- | --- | --- |
| CSRv1 | 271.32 | 638.77 |
| CSRv2-linear-StartEndTopk | 285.94 | 653.13 |
| CSRv2-DiverseMultiTopk | 501.18 | 1183.36 |
| CSRv2(anneal) | 274.15 | 642.65 |

These results demonstrate that 1) focusing on start and end Ks help a bit on addressing ultra-sparsity but there is still a large gap to our $k$-annealing; 2) better $k$ coverage with diverse multi-topk delivers larger gains, but it still underperforms $k$-annealing, while **introducing significant training overhead**. Therefore, we belive that $k$-annealing is more preferrable than these static multi-topK loss variants. It would be interesting to look deeper into their interplay in future work.

---

We hope that the newly added experiments address your remaining concerns. If you find it satisfactory, I respectivefully ask you to re-evaluate the rating. And please do not hesitate to let us know if there is any further questions!

---

### Author Response · Authors · 2025-11-28
**Response to Reviewer 6A2r and tchZ's  further comments**

Thank you very much for your additional comment. We fully agree that PQ quantization offers a more nuanced representation compared with binary quantization and therefore provides a complementary perspective that can improve the completeness of our quantization comparison.

Following your recommendation, we evaluated PQ on both baseline and CSRv2 embeddings with code budget 64. We use standard settings with 256 codewords per subspace and 8 subvectors, and for 128-bit codes into 16 subvectors. For CSRv2, we apply PQ quantization on topk=256's embedding.

As shown in the **table below**, **PQ does not outperform Binary Quantization (BQ) in this context.** For example, on CSRv2 (64-bit), BQ achieves **74.12** on classification, whereas PQ drops to **62.39**.

We attribute this performance gap to a fundamental **structural mismatch.** Standard PQ partitions vectors into independent subspaces with equal bit-budgets, implicitly *assuming a uniform distribution of semantic information*. However, MRL and CSRv2 embeddings are strictly **hierarchical**, concentrating "core" semantics in the early/sparse dimensions. Consequently, **PQ's uniform allocation strategy disrupts this hierarchy by inefficiently assigning equal capacity to both the highly informative prefix dimensions and the fine-grained tail dimensions**, resulting in suboptimal quantization.

*Performance comparison on CSRv2 and dense MRL in BQ/PQ settings under fixed bit size 64*
| Fixed Bit Size | Method | Quantization | Active Dim | Classification | Clustering | Retrieval |
|---|---|----|------|-----|------|---|
| 64   | CSRv2  | binary       | 64         | 74.12          | 44.53      | 40.28     |
| 64 | CSRv2 | PQ | 64 | 62.39 | 33.16 | 21.76 |
| 64      |   MRL   | binary | 64 | 64.48 | 40.04 | 27.61  |
| 64  | MRL | PQ | 64 | 58.37 | 37.18 | 22.04 |

Conversely, Binary Quantization **preserves the sign information of high-value dimensions directly**, offering superior compatibility with hierarchical representations. Thus, while PQ presents an interesting avenue, it necessitates adaptation to function effectively with MRL/CSRv2's hierarchical representations, which we leave for future work.

---

Thank you again for your thoughtful and constructive comments. We sincerely appreciate the time and expertise you have invested in reviewing our work and helping us strengthen the paper. We have carefully addressed all raised points and incorporated the corresponding revisions into the latest version. We hope that the updated manuscript adequately reflects your valuable feedback, and we remain grateful for your insightful guidance throughout this process.

---

### Author Response · Authors · 2025-11-30
**Rebuttal Summary**

Dear Program Chairs, Senior Area Chairs, Area Chairs, and Reviewers,

We sincerely appreciate the tremendous efforts of the Program Chairs, Senior Area Chairs, and especially the Area Chairs in coordinating the review process. We also extend our heartfelt thanks to all Reviewers for their constructive and detailed evaluations.

To facilitate the tracking of our revisions during the discussion phase, we summarize below the key concerns raised by the reviewers, the additional experiments and clarifications we have incorporated in the revised manuscript, and the current state of the discussion.

**Additional experiments.** A major point of concern across reviewers pertained to the breadth and depth of our experimental evaluation. Following their suggestions, we conducted a number of additional experiments and integrated the results into the updated submission:

1. Weighting strategies on top-k loss terms and alternatives to annealing in both MRL and CSRv2, now presented in Appendix G.3 and Appendix G.4. (Raised by Reviewer rU7X)
2. Sensitivity analysis of the k-schedule, including schedule shape, duration, and initialization, now presented in Appendix G.2. (Raised by Reviewer EiFR)
3. Broader baseline comparisons, including MRL-SAE and SPLADEv3, now presented in Appendix C.5 and Appendix H.5. (Raised by Reviewer rU7X)
4. CSRv2 generalization capability evaluation (zero-shot evaluation on GraphRAG Benchmark), now presented in Appendix E. (Raised by Reviewer rU7X)
5. Analysis of the superclass-separation capability of ultra-sparse representations, now presented in Appendix H.1. (Raised by Reviewer rU7X)
6. Quantized comparisons at fixed memory budgets, covering BF16, FP32, binary, and PQ quantization, now presented in Appendix H.2 and Appendix H.3. (Raised by Reviewers tchZ and 6A2r)

**Additional clarifications.** Beyond experimental extensions, we also added further discussions and clarifications to address conceptual and methodological questions raised by the reviewers:

1. Clarification of the novelty of each component in CSRv2, now highlighted in Section 3. (Raised by Reviewer EiFR)
2. Discussion on k-annealing sensitivity and comparisons with variant strategies for both MRL and CSRv2, now presented in Appendix G.2–G.4. (Raised by Reviewers EiFR and rU7X)
3. Efficiency and computational cost analysis, now presented in Appendix G.1. (Raised by Reviewer EiFR)
4. Clarification of CSRv2’s capability in generating top-k representations across all sparsity scales, now emphasized in Lines 362–364. (Raised by Reviewer rU7X)
5. Further discussion on the necessity and practical acceptance of ultrahigh sparsity, now presented in Appendix C.5. (Raised by Reviewer rU7X)
6. Interpretation of extremely sparse representations, especially in relation to superclass separation, now presented in Appendix H.1. (Raised by Reviewer rU7X)
7. Abstract restructuring to improve clarity of empirical contributions, now reflected in Lines 27–33. (Raised by Reviewer tchZ)
8. Additional details on MTEB experiment setups, now clarified in Lines 69–71 and Line 409. (Raised by Reviewer tchZ)
9. Clarifications on vector quantization (FP32/BF16/binary/PQ) under fixed bit budgets for both CSRv2 and MRL, now included in Appendix H.2 and Appendix H.3. (Raised by Reviewers tchZ and 6A2r)

**Writing improvements.** We additionally refined the presentation quality in response to reviewer feedback:

1. Clarified the inference procedure of CSR-style models for new readers, now included in Lines 367–369. (Raised by Reviewer 6A2r)

**Outcome.** Several reviewers acknowledged the updates and clarifications provided during the discussion phase:

1. Reviewers 6A2r and tchZ expressed strong appreciation of our response, while maintaining their initial ratings of 8.
2. Reviewer rU7X thanked us for the first-stage response, and we have addressed the two follow-up questions in our updated reply.
3. Reviewer EiFR has not participated in the discussion.

Overall, we believe the additional experiments, analyses, and clarifications have effectively resolved the concerns raised by all reviewers.

Thank you once again for your dedication to the community.

Best regards,

Authors

---

### Meta-Review · Area_Chair_zQdJ · 2025-12-08

**Summary:**

Novelty & source of gains: Is CSRv2 mainly “recipe” (k-annealing + SupCL + full FT) rather than a new principle; are the gains at ultra-sparsity coming from annealing or mostly supervision/finetuning?

Practicality & scope: Can one backbone truly serve all k at inference; how desirable/acceptable is ultra-sparsity given dead neurons and accuracy trade-offs; does reliance on supervised data limit generality?

Method details & sensitivity: How sensitive to the k-schedule (shape/length/init); could static multi-Top-K losses / weightings replace annealing; what about MRL weighting and MRL-SAE variants?

Evaluation breadth & efficiency: Need broader baselines (e.g., SPLADEv3, MRL-SAE), zero-shot generalization, quantization at fixed bit budgets (binary/PQ), and system costs (encoder/index, retrieval).

Clarity: Make CSR/CSRv2 inference clearer; tighten the abstract; document MTEB setups.

**Reviewer Concerns:**

Reviewer 6A2r (accept, asked for clarity & fixed-budget quantization)
Clarity (varying k, normalization) & Table-1 stars: Addressed.
Fixed-bit comparisons: Addressed. (64/128-bit comparisons across FP32/BF16/binary for CSRv2; later binary baselines for MRL under same budgets.)
PQ request: Mostly addressed. Authors added PQ at 64-bit (CSRv2 & MRL) and argued binary > PQ for their hierarchical embeddings; 6A2r still requested broader PQ (more budgets).

Reviewer tchZ (accept, minor clarity + quantization context)
Abstract specificity & MTEB setup details: Addressed.
Figure 1b baseline line (MRL): Addressed.
Quantization & compute-optimality (PQ/AVQ; AdANNS context): Partially addressed. (Added fixed-bit results and a discussion of CSRv2 + quantization; later a PQ result and reasoning, but no direct AdANNS (MRL+PQ) comparison.)

Reviewer rU7X (initial reject; conceptual/practical questions + follow-ups)
Single backbone for all k / multi-Top-K training: Addressed.
MRL weighting for small dims: Addressed. (Added MRL-reweight baseline.)
MRL-SAE variant: Addressed. (Added MRL-SAE; it underperforms vanilla SAE/CSR on embedding tasks.)
Reliance on supervision / domain generality: Addressed.
Ultra-sparsity desirability & dead neurons: Addressed.
What is learned at ultra-sparsity (superclass separation): Addressed.

Reviewer EiFR (borderline; novelty, source of gains, efficiency, sensitivity)
Novelty & where gains come from: Addressed.
k-schedule sensitivity: Addressed.
Efficiency metrics (encoder/index/end-to-end): Partially addressed. Reported encoder time on 1M corpus (near-identical to MRL) and retrieval speedups; index build time still not directly reported.

**Reviewer Scores:**

I cannot speculate on how reviewers would have modified their scores. 6A2r made clear that he didn't change the score and he was quite supportive of the paper. The authors entered into extensive discussions and provided important new data.

---

### Decision · Program_Chairs · 2026-01-26

Accept (Poster)